# Uncovering Intermediate Variables in Transformers using Circuit Probing

**Michael A. Lepori**[*]         **Thomas Serre**[†]         **Ellie Pavlick**[*]

{`michael_lepori, thomas_serre, ellie_pavlick`}@brown.edu

## Abstract

Neural network models have achieved high performance on a wide variety of complex tasks, but the algorithms that they implement are notoriously difficult to interpret. It is often necessary to hypothesize intermediate variables involved in a network's computation in order to understand these algorithms. For example, does a language model depend on particular syntactic properties when generating a sentence? Yet, existing analysis tools make it difficult to test hypotheses of this type. We propose a new analysis technique – *circuit probing* – that automatically uncovers low-level circuits that compute hypothesized intermediate variables. This enables causal analysis through targeted ablation at the level of model parameters. We apply this method to models trained on simple arithmetic tasks, demonstrating its effectiveness at (1) deciphering the algorithms that models have learned, (2) revealing modular structure within a model, and (3) tracking the development of circuits over training. Across these three experiments we demonstrate that circuit probing combines and extends the capabilities of existing methods, providing one unified approach for a variety of analyses. Finally, we demonstrate circuit probing on a real-world use case: uncovering circuits that are responsible for subject-verb agreement and reflexive anaphora in GPT2-Small and Medium.

## 1 Introduction

Transformer models are the workhorse of modern machine learning, driving breakthroughs in subfields as disparate as NLP (Devlin et al., 2018; Radford et al., 2019; Brown et al., 2020), computer vision (Dosovitskiy et al., 2020), and reinforcement learning (Chen et al., 2021). Despite their success, little is known about the algorithms that they learn to implement. This central problem has inspired a flurry of analysis and interpretability research trying to "open the black box" (Rogers et al., 2021; Elhage et al., 2021; Belinkov, 2022). Despite considerable effort, these models remain almost entirely opaque.

One challenge inherent to interpreting a model that succeeds at a complex task is that researchers often do not have a complete picture of the algorithm that they are attempting to uncover. However, they may be able to propose high-level causal variables that are constituents of such an algorithm. For example, one may intuit that computing the syntactic number of the subject noun of a sentence might be useful for language modeling (Chomsky, 1965; Linzen et al., 2016). This variable must be causally implicated in an algorithm that solves the language modeling task, as it constrains the rest of the sentence due to agreement rules (i.e. the syntactic number of the main verb must match the syntactic number of the subject). However, it leaves open infinite possibilities in which other variables influence the prediction of the next token. Though we focus on language modeling, this discussion applies more generally to any complex domain where neural networks are applied, from vision (Dosovitskiy et al., 2020) to astronomy (Ćiprijanović et al., 2020) to protein folding (Jumper et al., 2021).

---

[*]Department of Computer Science, Brown University
[†]Carney Institute for Brain Science, Brown University

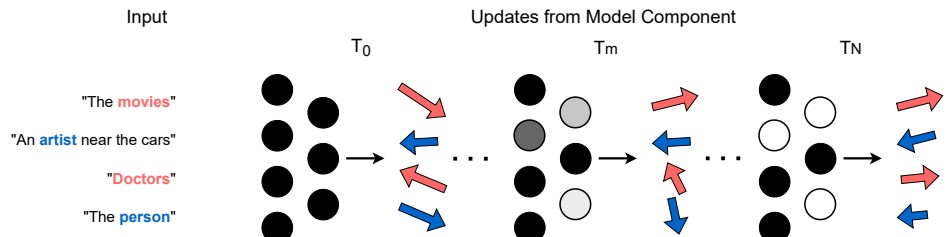

Figure 1: Schematic visualization of circuit probing for an intermediate variable representing the syntactic number of the subject of a sentence. Plural subjects are represented in red and singular subjects in blue. At step $T_0$, prior to training a binary mask, the model component (Attention block or MLP) produces residual stream updates (red and blue arrows; Elhage et al. (2021)) that are not partitioned by syntactic number (i.e. they will point in seemingly-random directions). Circuit probing optimizes a binary mask over model weights. By the end of mask optimization, the circuit will produce updates that are partitioned by syntactic number (i.e. point in one direction for singular subjects and another for plural subjects).

We propose *circuit probing* to enable the investigation of intermediate variables in Transformers (Vaswani et al., 2017). Circuit probing introduces a trainable binary mask over model weights that is optimized to uncover a circuit[1] that computes a high-level intermediate variable (if one exists). This technique combines the best aspects of standard probing and causal analysis methods, allowing researchers to (1) test whether high-level intermediate variables are represented by the model, (2) test whether they are *causally* implicated in the model behavior (rather than simply decodable from model representations), and (3) reveal the particular subset of model weights that compute it. We first demonstrate the benefits of circuit probing over existing methods using simple arithmetic tasks, showing that it is more faithful to the underlying model than existing methods (i.e. it only provides evidence in support of causal variables that are actually represented by the model) and reliably uncovers circuits that are causally implicated in model behavior. We then use circuit probing to analyze two syntactic phenomena on GPT2-Small and Medium (Radford et al., 2019), uncovering particular circuits responsible for subject-verb agreement and reflexive anaphora agreement[2].

## 2 Circuit Probing

When attempting to interpret a model that is performing some task, one must often hypothesize the existence of an intermediate variable that the model is computing. Circuit probing attempts to (1) measure whether this intermediate variable is computed and (2) identify the components of the model that are responsible for computing it (i.e. a circuit). Similar to prior work (Conmy et al., 2023; Cao et al., 2021), we attempt to to uncover model components by optimizing a binary mask over frozen model weights.

Recent work has shown that neural networks often exhibit structure at the level of subnetworks (Csordás et al., 2020; Lepori et al., 2023b; Hod et al., 2021). In light of this, we attempt to uncover circuits *within* individual attention and MLP blocks. These layers produce additive updates to the residual stream (Elhage et al., 2021). Intuitively, circuits that compute an intermediate variable should produce outputs that are partitioned according to that variable. For example, if a circuit is computing the syntactic number of the subject noun, then that circuit should produce outputs that fall into one of two equivalence classes, corresponding to singular subjects and plural subjects. Thus, we optimize a binary mask such that — if the variable is computed by the model within a particular layer — the outputs of that layer

---

[1]Similar to prior work Wang et al. (2022), we take the term"circuit" to mean "any subgraph of the computational graph describing a full model". We isolate circuits within individual attention and MLP blocks, and our circuits are composed of neurons. In general, circuit probing can be applied at any level of granularity, as long as the subgraph produces an update to the residual stream.

[2]We release our code at: https://github.com/mlepori1/Circuit_Probing. Circuit probing is implemented using the NeuroSurgeon package (Lepori et al., 2023a).

are clustered according to that variable. A natural optimization objective for this is the soft nearest neighbors loss (See Appendix D), a contrastive loss that minimizes the (cosine) distance between outputs from the same class, and maximizes the (cosine) distance between outputs from different classes (Salakhutdinov & Hinton, 2007; Frosst et al., 2019). Circuit probing is illustrated in Figure 1.

In this work, we mask at the neuron level (i.e. over columns in the matrices of the linear transformations that comprise MLP and attention blocks in the Transformer) in both MLP and attention blocks. We use continuous sparsification (Savarese et al., 2020), a pruning technique that anneals a soft mask into a discrete mask over training, to learn the binary mask (See Appendix C). We also train with $l_0$ regularization to encourage sparse binary masks. If this process is successful, it results in a sparse circuit within a model component that computes the hypothesized intermediate variable. See Algorithm 1 for pseudocode.

**Circuit Probing Evaluation:** We evaluate circuit probing in two ways: (1) We train a 1-nearest neighbor classifier on the output vectors produced by the discovered circuit, and then test this classifier on held-out data. If circuit probing succeeds in finding a circuit that computes a particular intermediate variable, then its output vectors will be partitioned by the possible labels of this variable, and we expect this classifier to achieve high performance. We employ a rather conservative strategy here, randomly sampling only 1 output vector for each label to train the nearest neighbors classifier.[3] See Algorithms 2 for pseudocode. (2) We ablate the discovered circuit (i.e. invert the learned binary mask) and analyze how the model's behavior changes.

**Causality:** This ablation is equivalent to asking the counterfactual question, "How does the model's output change if it does not compute a particular intermediate variable, $z$, in a particular block?", assuming that the circuit is *only* causally implicated in computing $z$. We note that this assumption is unlikely to be completely true, as neurons are typically polysemantic (Elhage et al., 2022). However, (1) we attempt to mitigate this by encouraging maximally sparse subnetworks using $L_0$ regularization and (2) prior work has suggested that sparse circuits may be surprisingly monosemantic (Hamblin et al., 2022).

---

**Algorithm 1** Training Step

**Input:** Model $M_\theta$, Batch $B$, Probe Layer $i$,
Mask Params $m$

1: $M_\theta = [M_{\theta 0 \ldots (i-1)}, M_{\theta i}, M_{\theta(i+1) \ldots N}]$
2: $B = \{x, y\}$
3: `Freeze`($\theta$)
4: $v \leftarrow M_{\theta i \odot m}(M_{\theta 0 \ldots (i-1)}(x))$
5: loss $\leftarrow$ `soft_neighbors`($v, y$)
6: loss $+= |m|$        $\triangleright$ $L_0$ loss
7: `backpropagate`(loss, m)

---

**Algorithm 2** Evaluation

**Input:** Model $M_\theta$, Train Data $D_{Train}$,
Test Data $D_{Test}$, Probe Layer $i$, Mask $m$

1: $M_\theta = [M_{\theta 0 \ldots (i-1)}, M_{\theta i}, M_{\theta(i+1) \ldots N}]$
2: $D_{Train} = \{x, y\}, D_{Test} = \{x'\}$
3: $v \leftarrow M_{\theta i \odot m}(M_{\theta 0 \ldots (i-1)}(x))$
4: $knn \leftarrow$ `train_knn`($v, y$)
5: $v' \leftarrow M_{\theta i \odot m}(M_{\theta 0 \ldots (i-1)}(x'))$
6: $\hat{y} \leftarrow$ `predict_knn`($knn, v'$)

---

## 3 Baselines

Circuit probing accomplishes two distinct goals: (1) it allows one to measure how well a circuit partitions inputs according to an intermediate variable (i.e. it provides a probing accuracy measurement), and (2) it allows one to perform causal analysis by ablating circuits. Prior methods accomplish either one of these goals, but not both. Thus, we use different baselines for each type of analysis. When analyzing probing accuracy, we compare circuit probing to "**vanilla probing**" (henceforth "probing") and **contrastive probing**. Probing involves learning a classifier to decode information about a hypothesized intermediate variable from model activations (Tenney et al., 2019; Hewitt & Manning, 2019; Ettinger, 2020; Li et al., 2022; Nanda et al., 2023). Prior work has demonstrated that probing oftentimes mischaracterizes the underlying computations performed by the model (Hewitt & Liang, 2019; Zhang & Bowman, 2018; Voita & Titov, 2020)). We empirically demonstrate that circuit

---

[3]We note that this is equivalent to a linear classifier whose weights are defined by the sampled vector for each class.

probing is more faithful to a model's computation than linear or nonlinear probing (See Sections 4.2, 4.3)[4]. Contrastive probing is an ablation of circuit probing where we train linear probes using a similar contrastive objective. We present results in Appendix R, and find that this method largely fails.

When performing causal analysis, we compare to three existing methods: **causal abstraction analysis**, **amnesic probing**, and **counterfactual embeddings**. Causal abstraction analysis intervenes on the activation vectors produced by Transformer layers to localize intermediate variables to particular vector subspaces (Geiger et al., 2021; 2023; Wu et al., 2023). However, it requires hypothesizing a complete causal graph – a high-level description of how inputs are mapped to predictions, including all interactions between intermediate variables. This is impossible for most real-world tasks on which we want to apply neural networks (such as language modeling, to which we apply circuit probing in Section 4.4). We empirically demonstrate the utility of boundless distributed alignment search (boundless DAS; Wu et al. (2023)), a state-of-the-art causal abstraction analysis technique, and show that circuit probing arrives at the same results (See Sections 4.1, 4.2, 4.3). Amnesic probing seeks to *erase* linearly-decodable information about an intermediate variable, and then observe the effect of this erasure on downstream behavior (Elazar et al., 2021). If this behavior is changed, then the intermediate variable is implied to exist within the original network. Though many techniques have been employed to erase linearly-decodable information (Ravfogel et al., 2020; 2022; Shao et al., 2023), we use the state-of-the-art LEACE method in our amnesic probing experiments (Belrose et al., 2023). We empirically demonstrate that circuit probing is more faithful to a model's computation than amnesic probing in Sections 4.2 and 4.3. Finally, counterfactual embeddings (Tucker et al., 2021) were introduced to enable causal analysis using particular probes. We find that counterfactual embeddings are fairly uninformative in our experiments (See Sections 4.1 and 4.2).

## 4 Experiments

We present four diverse experiments in order to illustrate the breadth of questions that circuit probing can help address. Experiments 1, 2, and 3 investigate toy models trained on simple arithmetic tasks, where full causal graphs are easy to construct. These experiments both (1) nuance or reproduce results from prior work and (2) compare circuit probing against existing analysis methods. We compare circuit probing against linear and nonlinear probing when assessing circuit probing's ability to decode intermediate variables and compare against amnesic probing, causal abstraction analysis, and counterfactual embeddings when performing causal analysis. Experiment 4 applies circuit probing to language models to demonstrate that the method scales to a more realistic model and setting[5].

### 4.1 Experiment 1: Deciphering Neural Network Algorithms

**Goal:** One of the central goals of interpretability research is to characterize the algorithms that models implement Olah (2022). This lofty goal is made substantially more tractable when we can adjudicate between two hypothesized alternatives. We demonstrate that all probing methods and most causal analysis methods produce converging results when characterizing the algorithm implemented by a model trained on a simple arithmetic task.

**Task:** We train a 1-layer GPT2 model to solve a task defined by the function $(a^2 - b^2)(\mathrm{mod}\ P)$, where P is set to 113, and $a$ and $b$ are input variables. The input sequences are of the form $[a, b, P]$, and the model is tasked with predicting the answer based on the output embedding of the final token. All inputs are symbolic – they are one-hot vector mappings to learnable embeddings. We exhaustively generate all possible data points with $a$ and $b$, taking values $0 - 112$. Note that this input-output mapping permits at least two possible solutions, because $a^2 - b^2 = (a + b) \times (a - b)$. We use circuit probing to determine which of the two alternative solutions the trained neural network adopts. Specifically, we search for the intermediate variables $a^2$, $(-1 * b^2)$, $(a + b)$, and $(a - b)$, all mod 113. We optimize binary masks using the output vectors from the attention and MLP blocks when they are operating on the $P$ token, which is where the final prediction is made.

---

[4]In the main text we probe the residual stream of the model, see Appendix Q for results of probing individual updates to the residual stream. Both methods give approximately the same results.

[5]See Appendix B for all data and hyperparameter details.

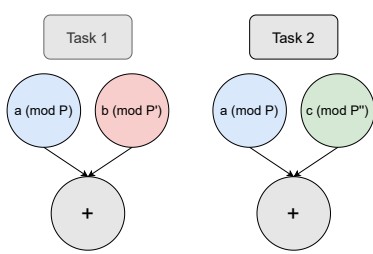

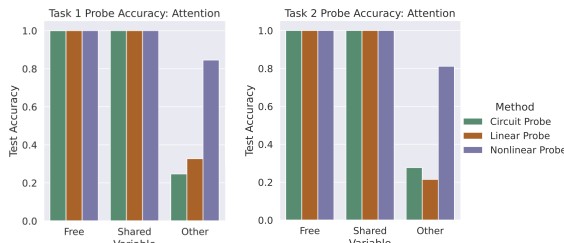

(a) Causal graphs representing the Multitask dataset in Experiment 2. Note that Task 1 and Task 2 share one intermediate variable (blue) and differ by one intermediate variable (red and green).

(b) Experiment 2 probing accuracy for circuit, linear, and nonlinear probes. We find that circuit probing and linear probing generally converge to find evidence of the Free and Shared variable for both tasks and no evidence of the irrelevant variable ("other"). On the other hand, nonlinear probing suggests that the irrelevant variable *is* represented.

**Probing:** We expect all methods to achieve high accuracy on *either* $a^2$ and $-1 * b^2$ **or** $a + b$ and $a - b$. This would adjudicate between the two alternative solutions to the arithmetic task. Indeed, we see that circuit probing, linear probing, and nonlinear probing[6] on the attention block all converge to ceiling accuracy for the variables in $a^2 - b^2$, and floor accuracy for the variables in $(a + b) \times (a - b)$ (See Appendix E). All methods converge toward more negative results from the MLP block, indicating that the intermediate variables are computed by the attention block (See Appendix E for MLP results).

**Causal Analysis:** We confirm that the circuits uncovered by circuit probing are causal in Appendix F. Amnesic probing weakly supports these conclusions (See Appendix G) and causal abstraction analysis strongly supports these conclusions (See Appendix H). On the other hand, we find that counterfactual embeddings fail to elicit counterfactual behavior in all cases, and thus do not provide evidence in either direction (See Appendix I). Finally, we run a transfer learning experiment, which behaviorally demonstrates that the model is representing the task as $a^2 - b^2$, rather than $(a + b) \times (a - b)$ (See Appendix J). Overall, our results demonstrate that circuit probing agrees with existing causal techniques when characterizing the algorithm implemented by a small model trained on a simple task.

## 4.2 Experiment 2: Modularity of Intermediate Variables

**Goal:** We now apply circuit probing to analyze the internal organization of Transformer models, which has been the subject of several recent studies (Lepori et al., 2023b; Csordás et al., 2020; Hod et al., 2021; Mittal et al., 2022). We show that circuit probing can be used to characterize whether computations are implemented in a modular and reusable manner within a Transformer, and that other methods fail to reveal such structure.

**Task:** We train a model on a simple multitask modular arithmetic task, represented by the causal graph in Figure 2a. We set $P = 29$, $P' = 31$, $P'' = 23$. The input sequences are of the form $[T, a, b, c, N]$, and the model is tasked with predicting the solution. $N$ is a separator token, and $T$ is a task token. For Task 1, we exhaustively generate all possible data points with $a$ and $b$, taking values $0 - 112$, and $c$ being a random token in the same range. Similarly, for Task 2. Note that both tasks share one intermediate variable (a "shared variable"), and each task has one intermediate variable that the other does not (a "free variable"). First, we probe for intermediate variables. We use circuit probing to probe each task individually, assessing which intermediate variables are computed when solving Task 1, and which are computed when solving Task 2. Specifically, we search for the intermediate variables $a(\text{mod P})$, $b(\text{mod P'})$, and $c(\text{mod P''})$. We optimize binary masks using the output vectors from the attention and MLP blocks when they are operating on the $N$ token, which is where the final prediction is made. Next, we investigate modularity directly. We hypothesize that the model implements a *reusable* computation for the shared variable $a(\text{mod P})$ — that the

---

[6](Non)linear probes are always trained to map from the residual stream of the final input token in the sequence (after the attention block and after the MLP block) to the value of the intermediate variable that we are attempting to decode.

same computation is used in both Task 1 and Task 2. Similarly, we hypothesize that the free variables are implemented *modularly* — that Task 1's free variable computation can be ablated without completely destroying performance on Task 2, and vice-versa.

**Probing:** We expect models to compute their free and shared variables ($a$(mod P) and $b$(mod P') for Task 1, and $a$(mod P) and $c$(mod P'') for Task 2), and not to compute the other variable. Our probing results for the attention block are shown in Figure 2b. Circuit, linear, and nonlinear probes decode free and shared variables with high accuracy, indicating that the relevant variables for a given task are computed in the attention block. However, nonlinear probing (and *only* nonlinear probing) reliably decodes the other variable. This accords with prior work questioning whether expressive probes accurately reflect the causal structure of neural networks (Voita & Titov, 2020; Zhang & Bowman, 2018).

**Causal Analysis:** Causal abstraction analysis agrees with circuit probing and linear probing (see Appendix K), causally implicating the free and shared variables for each task. On the other hand, counterfactual embeddings generated from the nonlinear probes once again fail to produce evidence that any variables are causal (see Appendix L). We conclude that counterfactual embeddings act more as adversarial examples for our trained nonlinear probe, rather than as meaningful counterfactual inputs to the model. Next, we perform a causal analysis to understand whether the model exhibits modularity. We expect that ablating the circuit computing the shared variable for *either* task should destroy performance across *both* tasks. On the other hand, we expect that ablating the circuit computing the free variable for Task 1 should destroy performance on Task 1 while having less of an effect on Task 2 performance (and vice-versa). In Appendix M, we analyze the morphology of the two free variable circuits returned by circuit probing and find that the circuits are only distinct in one tensor within the attention block. Thus, we only ablate circuit weights within that tensor. From Table 1, we see that ablating the circuit that computes the shared variable for *either* Task 1 or Task 2 destroys performance on both tasks. On the other hand, ablating the free variable in Task 1 destroys performance on Task 1, while maintaining some performance on Task 2. We observe the same trend for the free variable in Task 2. We run a similar analysis using amnesic probing and find that amnesic probing does not reveal this internal structure. We present results from amnesic probing on the residual stream after the attention block in Table 1. Overall, our results demonstrate that circuit probing is superior to competing techniques at characterizing how models structure their computations.

### 4.3 Experiment 3: Circuit Probing as a Progress Measure

**Goal:** Circuit probing allows us to gain insight into the training dynamics of Transformers at the level of intermediate variables. Recent work has shown that models may abruptly learn to generalize long after they overfit to the training data (Power et al., 2022). Despite this rather discontinuous switch from overfitting to generalization (often called *grokking*), Nanda et al. (2022) revealed that the circuit that computes the generalizable algorithm is formed continuously throughout training. While their work required reverse-engineering the entire algorithm to gain this insight into circuit formation, we reproduced their finding in a slightly different setting using circuit probing (which only requires us to hypothesize a high-level intermediate variable).

**Task:** We train a model on the task $(a^2 + b)$(mod P), with $P = 113$. The input sequences are of the form $[a, b, P]$, and the model is tasked with predicting the solution. Our model exhibits grokking on this task - it generalizes long after it overfits. Generalization performance increases rather slowly from epoch 0 until 10000, then rapidly climbs to near-perfect accuracy by epoch 17500. See Figure 3a. First, we probe for the development of the intermediate variable $a^2$ throughout training. Next, we perform a *selectivity analysis* on the trained model – we probe for a variable that is not causally implicated in model behavior ($b^2$) and verify that circuit probing does not decode this intermediate variable.

**Probing:** First, we investigate the development of the circuit computing $a^2$. We expect the performance of circuit probing to increase steadily throughout training, converging to a high value before the overall model generalizes. This finding would align with Nanda et al. (2022)'s finding that "circuit formation" occurs continuously, and that it completes before the overall model generalizes. We provide results from circuit, linear, and nonlinear probing on the attention block (See Appendix N for MLP results). From Figure 3b, we see that circuit probing accuracy increases throughout training, achieving >90% accuracy before epoch

| Method | Train Task | Variable | Task 1 Ablation Acc. | Task 2 Ablation Acc. |
|---|---|---|---|---|
| Circuit Probing | Task 1 | Shared | 3.7% | 3.7% |
| Circuit Probing | Task 1 | Free | 3.2% | **28.1%** |
| Circuit Probing | Task 2 | Shared | 3.6% | 3.6% |
| Circuit Probing | Task 2 | Free | **48%** | 4.8% |
| Amnesic Probing | Task 1 | Shared | 3.6% | 3.3% |
| Amnesic Probing | Task 1 | Free | 2.2% | **2.9%** |
| Amnesic Probing | Task 2 | Shared | 2.7% | 2.4% |
| Amnesic Probing | Task 2 | Free | **3.7%** | 4.5% |

Table 1: Experiment 2 circuit probing ablation and amnesic probing results. Both causal interventions require training (a binary mask for circuit probing, and an affine transformation for amnesic probing). "Train Task" refers to the task that was used for training these interventions. Targeted ablations of the circuits returned by circuit probing reveal a stark difference between the internal representations of the Free and Shared variables. A targeted ablation of the shared variable circuit destroys performance on both tasks, whereas a targeted ablation of Task 1's free variable harms performance on Task 1 far more (underlined) than Task 2 (bolded). The same is true in the opposite direction. We do not see these differences when performing amnesic probing.

10000. Thus, we can conclude that the circuit that allows the model to generalize is formed before the overall model's generalization behavior would imply. However, this circuit is not available from the outset and is developed throughout training. Linear and nonlinear probing tell a markedly different story, implying that the variable required to generalize was present nearly from the beginning of training. Next, we present results from our selectivity analysis. We expect our probing methods to achieve poor accuracy when probing for $b^2$, a variable that is not causally implicated in model behavior. At the end of training, circuit probing achieves 54.9% accuracy[7] at decoding $b^2$ from the fully trained model, whereas linear and nonlinear probing achieves 100% accuracy. These values are remarkably steady throughout training (See Figure 3c).

**Causal Analysis:** Causal abstraction analysis confirms that the $a^2$ intermediate variable gets progressively more relevant as training progresses, and that $b^2$ is never causally implicated in model behavior (See Appendix O). We find that ablating the circuits uncovered in the attention block by circuit probing for both $a^2$ *and* $b^2$ destroys model performance (See Appendix P). However, this is not a problem — the probing results make it clear that the circuit for $b^2$ is not successfully computing that intermediate variable in the first place, so the effect of ablating it is hard to predict. However, the linear classifier *does* decode both intermediate variables, and so amnesic probing is needed to clarify whether those variables are causal. We find that amnesic probing for $a^2$ incorrectly implies that this variable is causal right from the start, dropping test accuracy to near 0 throughout training. Amnesic probing for $b^2$ also incorrectly implies that this variable is causal throughout training, consistently dropping test accuracy (See Figure 3d). However, by the end of training, the effect of erasing information about $b^2$ is notably diminished. These results characterize an expected but important failure case of amnesic probing: $a^2$ and $b^2$ are linearly decodable from the *identity* of $a$ and $b$, so erasing all linearly-decodable information about either requires one to destroy the input. Overall these results demonstrate that circuit probing is more faithful to the underlying circuitry than existing methods.

### 4.4 Experiment 4: Circuit Probing in Language Models

**Goal:** The previous experiments focused on toy tasks in which a full causal graph could be specified. However, the reason that we are interested in developing interpretability tools is to analyze models that are used in practice on tasks where a causal graph cannot be constructed. Here, we use circuit probing to investigate how pretrained GPT2-Small and GPT2-Medium perform language modeling. In particular, we investigate two linguistic dependencies that rely on syntactic number: subject-verb agreement and reflexive anaphora.

---

[7]Random chance for circuit probing is 50%, as only two distinct integers in our dataset map to the same value of $b^2$(mod 113) and we are using a 1-nearest neighbor classifier.

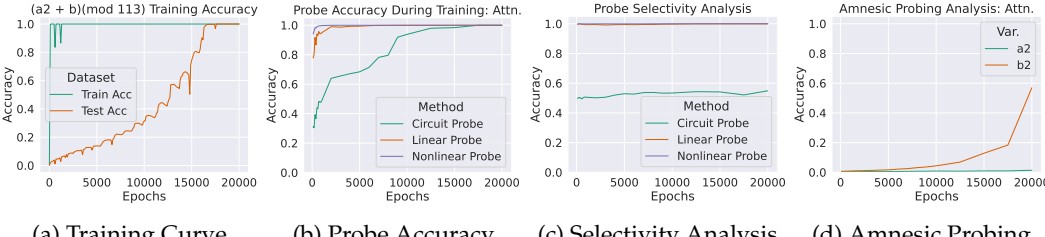

(a) Training Curve  (b) Probe Accuracy  (c) Selectivity Analysis  (d) Amnesic Probing

Figure 3: **(a)** We see generalization long after overfitting. **(b)** Probing for $a^2$. Linear and nonlinear probing converge to perfect accuracy very early in training, while circuit probing reveals that the circuit for $a^2$ is formed gradually through training. **(c)** Probing results for $b^2$, which is *not* causally implicated in the task $a^2 + b$. Circuit probing reveals that this variable is not represented at any point during training, whereas other methods imply that it is represented from the start of training. **(d)** Amnesic Probing incorrectly implies that (1) $a^2$ is causally implicated from the start, and (2) $b^2$ is causally implicated throughout training.

Subject-verb agreement refers to the English-language phenomenon where the subject of a sentence must agree with the main verb of a sentence in syntactic number. For example: *The **keys** are on the table* is grammatical, whereas *The **keys** is on the table* is ungrammatical. We hypothesize that an intermediate variable representing the syntactic number of the subject noun is computed when predicting the main verb of a sentence. Reflexive Anaphora ensures that reflexive pronouns agree with their referents. For example: *the **consultants** injured themselves* is grammatical, and *the **consultants** injured herself* is ungrammatical. We hypothesize that an intermediate variable representing the syntactic number of the referent is computed when predicting a reflexive pronoun.

**Task:** We use the templates from Marvin & Linzen (2018) to generate sentence prefixes, where the continuation of the prefixes are likely to be either a main verb (when studying subject-verb agreement) or a reflexive pronoun (when studying reflexive anaphora). See Appendix S for example prefixes. We run circuit probing on sentences containing one *distractor*, which is a non-subject or non-referent noun (e.g. "cabinet" in *the **keys** to the cabinet are on the table*). For each phenomenon, we run circuit probing on the last token of sentence prefixes to uncover the circuit that computes the syntactic number of the subject noun or referent. We then ablate the discovered circuit and evaluate the model's ability to continue held-out sentence prefixes grammatically. Specifically, we assess whether the model is more likely to predict tokens that are consistent or inconsistent with the syntactic number of the subject/referent. For subject-verb agreement, we inspect the logits for the tokens *is* and *are* – if the logit for *is* is higher than the logit for *are* when the subject is singular (e.g. *The officer...*), then we consider the model to have succeeded on that sentence prefix. For reflexive anaphora, we run the analysis twice, once comparing the logits of *herself* and *themselves*, and again comparing the logits of *himself* and *themselves*. We evaluate models on IID sentences with one distractor noun and on an OOD dataset of sentences that contain two distractor nouns (See Appendix S for examples). We hypothesize that the same circuit computes the relevant linguistic dependency for both sentence structures. Even if we recover positive results from this analysis, it is possible that we are simply destroying the entire model, rather than ablating a specialized circuit. As a control, we sample 5 random subnetworks from the complement set of neurons that are in our circuit and rerun the ablation analysis. Randomly-sampled subnetworks always contain the same number of neurons as our circuit.

**Probing:** See Appendix T.1 and U for an investigation of circuit probing accuracy. Generally, we find that most attention layers can compute the correct syntactic number, but that MLPs only begin to achieve good performance in the middle layers of GPT2-Small and Medium. See Appendix Y for linear probe accuracy for all GPT2 settings.

**Causal Analysis:** For both syntactic dependencies, we expect that ablating the discovered circuit will render the model worse at distinguishing the syntactic number of the subject/referent. We expect that ablating random subnetworks should not harm model performance on any dataset. We present results from GPT2-Small in the main body, and

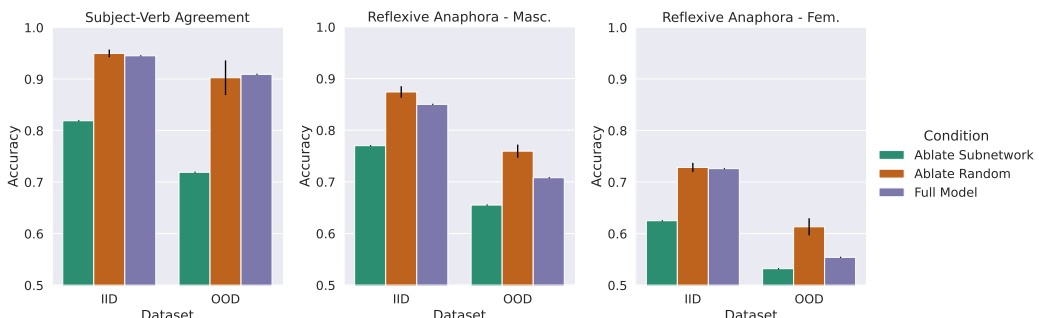

Figure 4: Experiment 4 GPT2-Small ablation results on layer 6's attention block. Across both Subject-Verb Agreement (Left) and Reflexive Anaphora evaluated using the masculine (Middle) and feminine (Right) pronoun, we see that ablating the discovered circuit renders the model worse at distinguishing syntactic number. Ablating randomly sampled subnetworks has does not hurt the model's ability to distinguish singular and plural subjects/referents.

present GPT2-Medium results in Appendix U[8]. For both phenomena, we find that the dependency is computed in layer 6's attention block[9]. Ablating the circuit returned by circuit probing drops performance substantially for IID and OOD datasets for both phenomena, while ablating random subnetworks does not impact model performance (See Figure 4). Other blocks do not exhibit this characteristic pattern as strongly (See Appendix T.2). This accords with prior work suggesting that syntactic dependencies are represented in middle layers of Transformers (Tenney et al., 2019; Vig & Belinkov, 2019). See Appendix V for an analysis of circuit overlap, Appendix W for qualitative results, and Appendix X for results demonstrating that syntactic number of individual tokens is computed earlier in the model.

## 5 Discussion

**Related Work:** Circuit probing is related to recent efforts in *mechanistic interpretability* — a burgeoning field that attempts to reverse-engineer neural network algorithms. Through substantial manual effort, researchers have uncovered the algorithms that both toy models (Olsson et al., 2022; Nanda et al., 2022; Chughtai et al., 2023) and more realistic models (Wang et al., 2022; Hanna et al., 2023; Merullo et al., 2023) are implementing. More broadly, there has been substantial work analyzing the syntactic (Linzen & Baroni, 2021; Goldberg, 2019; Tenney et al., 2018; McCoy et al., 2018) and semantic capabilities Pavlick (2022; 2023); Yu & Ettinger (2020); Hupkes et al. (2020); Dziri et al. (2023) of language models. Circuit probing is also related to work that attempts to decompose models into functional subnetworks Csordás et al. (2020); Hamblin et al. (2022); Lepori et al. (2023b); Zhang et al. (2021); Panigrahi et al. (2023); Hod et al. (2021); Cao et al. (2021). The success of circuit probing is further evidence that subnetworks are a useful lens through which to analyze models.

**Conclusion:** We introduce circuit probing, a novel method for uncovering low-level circuits that compute high-level intermediate variables. Through four experiments, we have shown that one can gain insights into the underlying algorithms the model is implementing, how these algorithms are structured within the model, and how they develop throughout training. Circuit probing combines and extends the capabilities of existing methods and outperforms them in several settings. However, it is currently unknown how multiple circuits compose within a given block to create one additive update to the residual stream, so one cannot replace individual variables to perform counterfactual interventions. Future work might seek to understand how circuits compose with one another for this purpose.

---

[8]Our results on GPT2-Medium are largely reproduce what we find in GPT2-Small for reflexive anaphora. We fail to identify a circuit that computes subject-verb agreement dependencies.

[9]For reflexive anaphora, we note that models achieves higher accuracy when predicting the masculine pronoun. This is evidence of gender bias in language models, which has been well-documented elsewhere (Marvin & Linzen, 2018; May et al., 2019; Rudinger et al., 2018; Weidinger et al., 2021).

## 6 Ethics Statement

Circuit probing can be used to uncover computations that a neural network is performing. This may have future implications for bias, fairness, and safety of neural network models. However, we emphasize that the current iteration of circuit probing should not be used in isolation to assess models for social biases in real-world systems. Circuit probing can provide positive evidence that a computation is implemented, but cannot yet be used to provide evidence that a computation is definitely *not* implemented in a real-world system.

## 7 Acknowledgments

The authors thank the members of the Language Understanding and Representation Lab and Serre Lab for their valuable feedback on this project and Rachel Goepner for proofreading the manuscript. This project was supported by ONR grant (N00014-24-1-2026). The computing hardware was supported by NIH Office of the Director grant S10OD025181 via the Center for Computation and Visualization (CCV) at Brown University.

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

## A    Reproducibility Statement

To foster reproducibility, we provide details on model training and hyperparameters in Appendices B and C. We provide details of our experimental design for all experiments throughout the main text, as well as in our appendices. Additionally, we provide a high-level explanation of the circuit probing algorithm in Section 2, and provide details on the loss function in Appendix D. Finally, we make our code publicly available.

## B    Data and Hyperparameter Details

All 1-layer GPT2 models have 4 attention heads, an embedding size of 128, and an MLP dimension of 512. We use the Adam optimizer (Kingma & Ba, 2014) with a learning rate of 0.001 and train with weight decay.

**Experiment 1**    We train a 1-layer GPT2 model on 60% of all possible datapoints for this task, leaving the other 40% held out as a test set. For circuit probing, we train our mask on all examples from the train set (See Appendix C for those training hyperparameters) using a batch size of 500, and use updates generated from train set examples to train the 1-nearest neighbors classifier. We evaluate on the held-out test set.

For linear and nonlinear probing, we train for 100 epochs, using a learning rate of 0.1. We use this same learning rate for generating counterfactual embeddings. Our nonlinear classifier uses ReLU nonlinearity, and has a hidden size of 256.

For Boundless DAS, we train for 250 epochs, with 2500 training examples using the Adam optimizer with a learning rate of 0.01.

For transfer experiments, we finetune on 60% of each dataset, using the Adam optimizer with a learning rate of 0.001. We train with weight decay. We train for fewer epochs with $a^2$ because the models converge very early in training.

**Experiment 2**    All details are the same as in Experiment 1.

**Experiment 3**    We train a 1-layer GPT2 model on 33.3% of all possible datapoints for this task, which gives us the grokking behavior that we wish to investigate. All other details are the same as Experiment 1.

**Experiment 4**    We train circuit probing on 2000 examples for each dataset and test on 1000 examples, otherwise all details are the same.

## C    Continuous Sparsification Details

Continuous sparsification enables us to train binary masks over model weights. Our loss function is defined as:

$$\min_{m_i \in \{0,1\}} L_{soft\_neighbors}(C_{\theta \odot m_i}) + \lambda ||m|| \tag{1}$$

Where $m$ is our binary mask, $C_\theta$ is a model component, with weights $\theta$, and $||m||$ is our $l_0$ regularizer, and $L_{soft\_neighbors}$ is the soft nearest neighbors loss described in Appendix D

Typically, optimizing such a binary mask is intractable, given the combinatorial nature of a discrete binary mask over a large parameter space. Instead, continuous sparsification reparameterizes the loss function by introducing another variable, $s \in \mathbb{R}^d$:

$$\min_{s_i \in \mathbb{R}^d} L_{soft\_neighbors}(C_{\theta \odot \sigma(\beta \cdot s_i)} + \lambda ||\sigma(\beta \cdot s_i)||_1 \tag{2}$$

In Equation 2, $\sigma$ is the sigmoid function, applied elementwise, and $\beta$ is a temperature parameter. During training, $\beta$ is increased after each epoch according to an exponential

schedule to a large value $\beta_{max}$. Note that, as $\beta \to \infty$, $\sigma(\beta \cdot s_i) \to H(s_i)$, where $H(s_i)$ is the *heaviside function*.

$$H(s) = \left\{ \begin{array}{l} 0, s < 0 \\ 1, s > 0 \end{array} \right\} \tag{3}$$

Thus, during training, we interpolate between a soft mask ($\sigma$) and a discrete mask ($H$). During inference, we simply substitute $\sigma(\beta_{max} \cdot s_i))$ for $H(s_i)$. Notably, we apply continuous sparsification to a frozen model in an attempt to reveal the internal structure of this model. In contrast, the original work introduced continuous sparsification in the context of model pruning and jointly trained $\theta$ and $s$.

For all experiments, we train binary masks with the Adam optimizer (Kingma & Ba, 2014), with a learning rate of 0.001. We fix $\beta_{max} = 200$, initialize the mask parameters to 0, and train for 90 epochs. The $\lambda$ parameter must scale with the number of parameters per layer. For all 1-layer GPT2 experiments, we set $\lambda = 1E - 6$. For GPT2-Small and Medium, we set $\lambda = 1E - 7$ and $1E - 8$, respectively.

## D    Soft Nearest Neighbors Loss

Given input embeddings $x$ to a model component $C$ and intermediate variable labels $y$, in a batch with $b$ samples, Equation 4 defines the full optimization objective. $\lambda$ is a hyperparameter that scales the $l_0$ regularization strength. Intuitively, this loss function pushes members of the same class towards each other, according to some distance metric, and members of different classes far from each other. Concretely, this partitions the output space of transformer layers into equivalence classes defined by the variable that we are searching for.

$$min_{m_n \in \{0,1\}} - \frac{1}{b} \left( \sum_{i \in 1..b} \frac{\sum_{\substack{j \in 1...b, \\ j \neq i, \\ y_i = y_j}} e^{cosine\_dist(C_{\theta \odot m}(x_i), C_{\theta \odot m}(x_j))}}{\sum_{\substack{k \in 1...b, \\ k \neq i}} e^{cosine\_dist(C_{\theta \odot m}(x_i), C_{\theta \odot m}(x_k))}} \right) + \lambda \sum_{n \in 1...|m|} m_n \tag{4}$$

## E    Experiment 1: Probing Results

In Table 2, we see that all methods converge to show that the model is computing $a^2 - b^2$, rather than $(a + b) \times (a - b)$.

In Figure 5, we present results from Experiment 1 for all probing methods on the MLP block. First, we note that all methods perform worse at decoding $a^2$ and $-1 * b^2$. We note that chance accuracy for circuit probing is effectively 50%, whereas chance for probing methods is 0.8% (1 out of 113). This is because circuit probing results are generated by a 1-nearest neighbors classifier trained on the outputs of the MLP block after masking. For the variable $a^2$ and $-1 * b^2$, there are only two distinct integers that map to the same value of that variable (i.e. $4^2 (mod\ 113) = 111^2 (mod\ 113) = 4$). Because we are training the classifier with 1 vector per variable label, 50% of the underlying integers are represented in the 1-NN training set. Thus, circuit probing accuracy of 50% means that the block is merely decoding the identity of the underlying token rather than meaningfully computing an intermediate variable.

## F    Experiment 1: Circuit Probing Causal Analysis Results

Table 3 contains the results from running a causal analysis on the circuits discovered in Experiment 1. We note two things: (1) ablating the circuits for $a^2$ or $-1 * b^2$ destroys model performance, and (2) circuit probing returns empty subnetworks for $a + b$ and $a - b$. This is

| Method | $a^2$ | $-1 * b^2$ | $a + b$ | $a - b$ |
|---|---|---|---|---|
| Circuit | 99% | 99% | 1% | 1% |
| Linear | 100% | 100% | 0% | 0% |
| Nonlinear | 100% | 100% | 1% | 1% |

Table 2: Experiment 1 probing accuracy for circuit, linear, and nonlinear probes. All methods converge to the conclusion that the model is representing $a^2 - b^2$, rather than $(a + b) * (a - b)$.

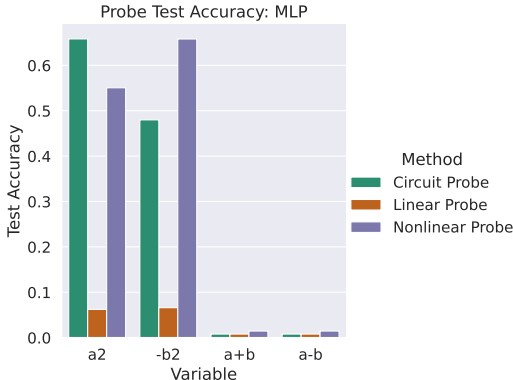

Figure 5: MLP probe accuracy for Experiment 1. All methods decode $a^2$ and $-1 * b^2$ worse in the MLP than in the attention block. Note that chance accuracy for circuit probing is effectively 50%.

a useful feature of using $l_0$ regularization when training binary masks – if there is no signal for a given variable, circuit probing is encouraged to return a maximally sparse (i.e. empty) subnetwork.

## G   Experiment 1: Amnesic Probing Results

In Table 4, we provide results from running amnesic probing using LEACE (Belrose et al., 2023) on the model in Experiment 1. LEACE requires fitting an affine transformation over an embedding to surgically erase all linearly-decodable information about an intermediate variable. We fit this transformation on the residual streams of the $P$ token taken from the test set. We then erase the linearly-decodable information from these same embeddings, patch them back into the model, and compute its post-intervention test accuracy. We find that erasing the intermediate variable $a^2$ or $-b^2$ after the attention block completely destroys model performance. Erasing $a + b$ or $a - b$ *also* harms performance, though not quite as much. This weakly supports the conclusions from circuit probing, but raises important

| Variable | Full Model Test Acc. | Ablated Test Acc. | % Parameters in Circuit |
|---|---|---|---|
| $a^2$ | 100% | 0.8% | 53.3% |
| $-b^2$ | 100% | 0.9% | 53.5% |
| $a + b$ | 100% | 100% | 0% |
| $a - b$ | 100% | 100% | 0% |

Table 3: Experiment 1 task performance after ablating the circuit returned by circuit probing. We see that ablating the circuit responsible for either $a^2$ or $-b^2$ destroys test accuracy. However, we see that circuit probing returns an empty circuit for both $a + b$ and $a - b$, due to $l_0$ regularization. Thus, ablating this empty circuit has no effect.

| Component | $a^2$ | $-b^2$ | $a+b$ | $a-b$ |
|---|---|---|---|---|
| Attn. | 1.8% | 1.8% | 35.3% | 35.2% |
| MLP | 100% | 99.9% | 99.3% | 99.4% |

Table 4: Amnesic Probing results for Experiment 1. We note a greater performance drop when erasing $a^2$ or $-b^2$ than $a+b$ or $a-b$ in the residual stream after the attention block.

| Component | $a^2$ | $-b^2$ | $a+b$ | $a-b$ |
|---|---|---|---|---|
| Attn. | 98% | 99% | 1% | 1% |
| MLP | 2% | 2% | 2% | 2% |

| Component | Task | $a$ (mod P) | $b$ (mod P') | c (mod P'') |
|---|---|---|---|---|
| Attn. | 1 | 93% | 93% | - |
| MLP | 1 | 3% | 3% | - |
| Attn. | 2 | 93% | - | 94% |
| MLP | 2 | 4% | - | 4% |

Table 5: Causal abstraction analysis results for Experiment 1 (top) and Experiment 2 (bottom). Using boundless distributed alignment search, we reveal that the attention blocks in both models contain the same causal intermediate variables that circuit probing discovers.

questions about the utility of LEACE when linear probes perform poorly (as is the case for $a+b$ or $a-b$, see Section 4.1).

## H  Experiment 1: Causal Abstraction Analysis Results

In Table 5, we provide results from running Causal abstraction analysis using Boundless DAS (Wu et al., 2023) on both Experiment 1 and Experiment 2. Causal abstraction analysis creates interventions on model representations in order to elicit counterfactual behavior in the downstream model. For example, in the case of $a^2 - b^2$, the model might intervene to change the value of $a^2$ to $a'^2$. The intervention is considered successful if the overall model outputs the answer to $a'^2 - b^2$. Causal abstraction analysis reports statistics in terms of the success of its counterfactual embeddings, rather than its ability to decode model representations.

These results support the results generated by circuit probing in Experiment 1. Causal abstraction analysis reveals evidence for $a^2$ and $-1 * b^2$, but not $a+b$ and $a-b$.

## I  Experiment 1: Counterfactual Embeddings

We present counterfactual embedding results from Experiment 1 in Table 6. Counterfactual embeddings are embeddings that are optimized to fool a probing classifier. Formally, given a probe, $P_\theta$ trained to decode an intermediate variable, $V$, consider a residual stream state $e$ such that $P_\theta(e) = V_i$. We freeze $P_\theta$ and optimize $e$ such that $P_\theta(e') = V_j$. If $P_\theta$ is decoding information that is causally implicated in the underlying model, then replacing $e$ with $e'$ should change the output to the output one would expect from setting variable $V$ to $V_j$. We report counterfactual embedding success – the percentage of embeddings that are successfully optimized to fool the probing classifier. We see that all linear probing classifiers can be fooled by counterfactual embeddings. We see that nonlinear classifiers can be fooled by counterfactual embeddings only for $a^2$ and $-1 * b^2$. Recall that all classifiers performed poorly at decoding $a+b$ and $a-b$.

Next, we analyze counterfactual behavior success – the percent of counterfactual embeddings that actually elicit counterfactual behavior in the overall model. We see that all sets of counterfactual embeddings fail to elicit counterfactual behavior. Taken in isolation, one

| Counterfactual Embedding Success | | | | |
|---|---|---|---|---|
| Probe | $a^2$ | $-b^2$ | $a+b$ | $a-b$ |
| Linear | 100% | 100% | 100% | 100% |
| Nonlinear | 100% | 100% | 1% | 1% |
| Counterfactual Behavior Success | | | | |
| Probe | $a^2$ | $-b^2$ | $a+b$ | $a-b$ |
| Linear | 1% | 1% | 1% | 1% |
| Nonlinear | 1% | 1% | 0% | 0% |

Table 6: Experiment 1 counterfactual embedding results. (Top) Counterfactual embedding success – the percent of examples where the counterfactual optimization procedure creates an example that changes probe outputs to a particular class. We see that this optimization process largely succeeds, except for $a+b$ and $a-b$ in nonlinear probes. (Bottom) Counterfactual behavior success – the percent of counterfactual embeddings that elicit counterfactual behavior in the model. We see very poor performance on this metric, indicating that counterfactual embeddings are not producing the expected behavioral outcomes.

might conclude that these probes are not decoding causally-relevant information, and thus that models are not actually computing $a^2$ and $-1 * b^2$. However, given the success of every other analysis technique at causally implicating $a^2$ and $-1 * b^2$, we may instead conclude that counterfactual embeddings are acting as adversarial examples to the probing classifier, and are destroying the embedding with respect to the underlying model.

## J  Experiment 1: Transfer Learning

To further confirm our findings in Experiment 1, we analyze whether training on $a^2 - b^2$ confers any benefits when finetuning on different tasks. In particular, we finetune the GPT2 model on a task defined by $a^2(\text{mod } 113)$, and separately on a task defined by $a+b(\text{mod } 113)$. If the model is solving the task using $a^2 - b^2$, we expect that finetuning should help the model solve $a^2$ faster than training a randomly initialized model, because the model already represents the variable necessary to solve the finetuning task. Similarly, we expect the finetuning to $a+b$ will be slower than training a randomly initialized model, because the model represents variables that are explicitly not useful for solving the finetuning task. From Figure 6, that is exactly what we see.

## K  Experiment 2: Causal Abstraction Analysis Results

In Experiment 2, causal abstraction analysis reveals evidence that the model is using variables $a(\text{mod P})$ and $b(\text{mod P'})$ when solving Task 1, and $a(\text{mod P})$ and $c(\text{mod P''})$ when solving Task 2 (See Table 5). Unfortunately, there is no causal model that involves $c(\text{mod P'})$ when solving Task 1, and $b(\text{mod P''})$ when solving Task 2, and so it does not make sense to run causal abstraction analysis in this setting.

## L  Experiment 2: Counterfactual Embedding Analysis

Here, we run a causal analysis of the nonlinear probes using counterfactual embeddings. Counterfactual embeddings are designed to reveal whether probes are reflecting information that is causally implicated in model behavior. We summarize the relevant details of their technique here but defer to Tucker et al. (2021) for a full treatment. Given a probe, $P_\theta$ trained to decode an intermediate variable, $V$, consider a residual stream state $e$ such that $P_\theta(e) = V_i$. We freeze $P_\theta$ and optimize $e$ such that $P_\theta(e') = V_j$. If $P_\theta$ is decoding information that is causally implicated in the underlying model, then replacing $e$ with $e'$ should change the

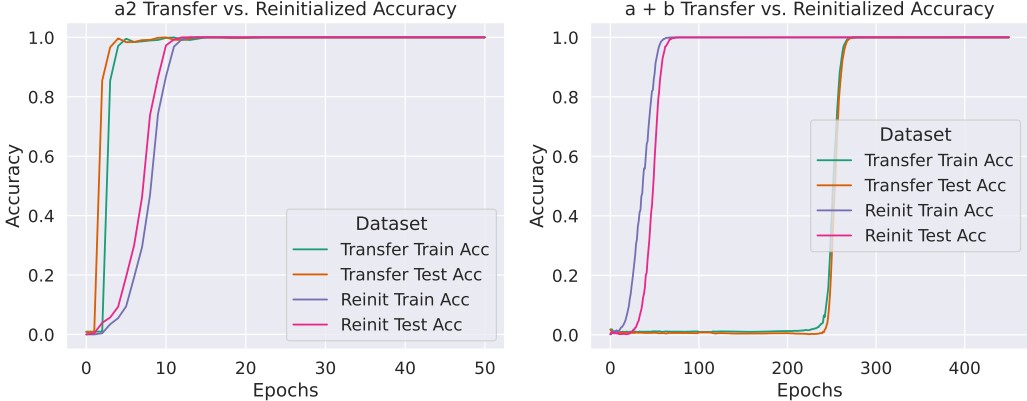

Figure 6: (Left) Transfer performance for $a^2$. We see that pretraining on $a^2 - b^2$ confers a benefit to the model when finetuning on $a^2$. (Right) Transfer performance for $a + b$. We see that pretraining on $a^2 - b^2$ is a detriment when finetuning on $a + b$.

| Task | Probe Var. | CE Success | Model Acc. | Counterfactual Acc. |
|------|-----------|-----------|-----------|---------------------|
| 1 | Free | 100% | 2.1% | 2.2% |
| 1 | Other | 100% | 1.8% | 1.5% |
| 1 | Shared | 100% | 2.5% | 2.4% |
| 2 | Free | 100% | 2.1% | 2.1% |
| 2 | Other | 100% | 2.3% | 2.1% |
| 2 | Shared | 100% | 2.2% | 3.0% |

Table 7: Experiment 2 counterfactual embedding results. Counterfactual embeddings that created by "free" and "shared" variable probes should result in a different prediction being made in the overall model. Counterfactual embeddings created by the "other" variable probe should have no effect on the overall model's prediction. We see that this does not happen. Though all counterfactual embeddings succeed at changing the probe prediction (CE Success), they also all change the overall model prediction (Model Acc.), but not to the correct counterfactual answer (Counterfactual Acc.).

output according to the counterfactual variable. If this information is not causally implicated in the underlying model, then replacing $e$ with $e'$ should have no effect, preserving the original model output.

We generate counterfactual embeddings for both tasks using the nonlinear probes trained to decode the "free" and "other" variables from the residual stream after the attention block. We record the percentage of examples whose output is maintained before and after substituting the counterfactual embedding (the "Model Accuracy"). We expect this value to be high for the "other" variable, and low for the "free" and "shared" variables, indicating that the features that the nonlinear probe uses to decode the "other" variable are *not* causal. Conversely, we record the percentage of examples whose output is changed according to the counterfactual variable (the "Counterfactual Accuracy"). We expect this value to be high for the "free" and "shared" variables and low for the "other" variable. This would indicate that only the "free" and "shared" variables are causal.

From Table 7, we see that both Model Accuracy and Counterfactual Accuracy drop to near-zero after patching in counterfactual embeddings for *either* "free", "shared", or "other" variables. This suggests that counterfactual embeddings act more as adversarial examples to the probe, rather than providing useful information about causally-relevant variables.

Free Variable Circuit Overlap

Figure 7: Experiment 2: Visualizing the distribution of circuits throughout the tensors comprising an attention block. We see that the circuits computing the free variables for Task 1 and Task 2 almost completely overlap in the c_proj tensor, but are mostly distinct in the c_attn tensor.

## M    Experiment 2: Circuit Overlap Analysis

From Figure 7, we see that the two circuits computing the free variables in Task 1 and Task 2 are largely distinct in `attn.c_attn`, but nearly completely overlapping in `attn.c_proj`. With this insight, we ablate circuit parameters just within `attn.c_attn`. We can also visualize the distribution of circuits through attention heads. In Figure 8, we see that circuit probing recovers structures that exist between particular attention heads (i.e. no single attention head is *fully* devoted to an intermediate variable), but also partially localizes the two Free variables into specific heads (head 1 for Task 1, head 2 for Task 2).

## N    Experiment 3: MLP Probe Accuracy

Here we present the linear, nonlinear, and circuit probe accuracy on the MLP block throughout training for Experiment 3. In Figure 9, we see very messy results, further reinforcing that the intermediate variable $a^2$ is computed in the attention block. We also present amnesic probe accuracy after erasing each intermediate variable from the residual stream after the MLP block in Experiment 3. In Figure 10, we see that test accuracy monotonically increases throughout training, generally reflecting the test accuracy of the underlying model. This indicates that amnesic probing does not erase any important information

## O    Experiment 3: Causal Abstraction Analysis

We present results from running causal abstraction analysis throughout training for the variables $a^2$ and $b^2$. $a^2$ should be causally implicated in model behavior by the end of training, as the model generalizes to unseen data according to the function $a^2 + b$. $b^2$ should never be causal, as it is not relevant to the task. We see the expected patterns in Figure 11.

## P    Experiment 3: Circuit Probing Causal Analysis

Figure 12 provides our results from ablating the circuits uncovered by circuit probing throughout training. Probing results already indicate that the circuit uncovered for $b^2$ fails

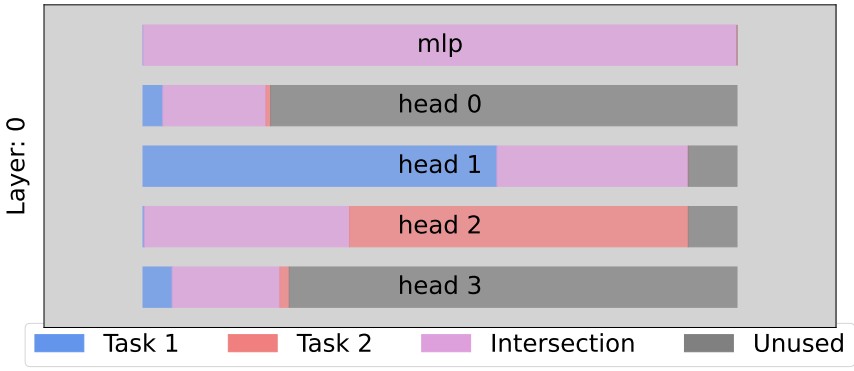

Figure 8: Circuit probing recovers some elements of known structure within Transformers. In particular, we see that Head 1 largely computes the Free variable in Task 1, and Head 2 largely computes the Free variable in Task 2. However, we also note that circuits extend beyond individual attention heads.

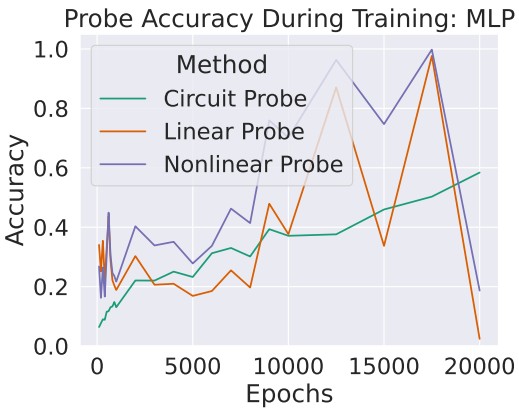

Figure 9: Experiment 3: MLP Probing results. We see chaotic results from all probes, indicating that the intermediate variable $a^2$ is not computed in the MLP block.

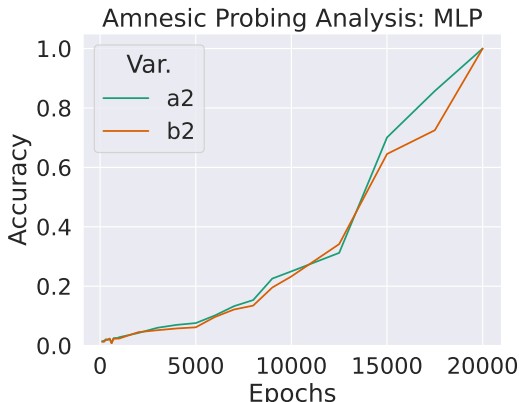

Figure 10: Experiment 3: MLP Amnesic Probing results. We see that amnesic probing has little impact on test accuracy throughout training.

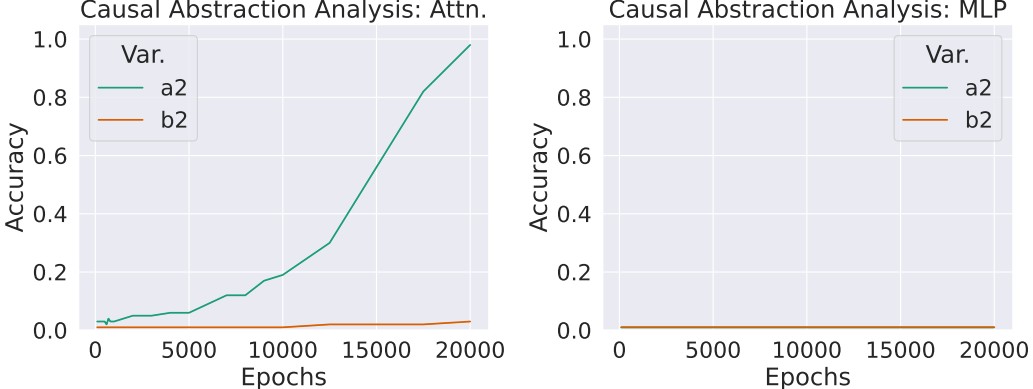

Figure 11: Experiment 3: Causal abstraction analysis results for both $a^2$ and $b^2$. $a^2$ becomes causal in the attention block throughout training, indicating a switch from memorization to generalization. $b^2$ never becomes causal, as expected.

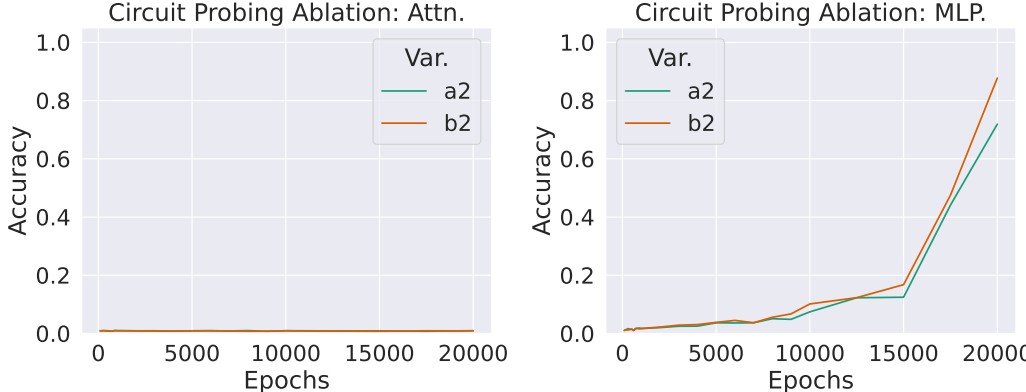

Figure 12: Experiment 3: Ablating the circuits discovered by circuit probing in the attention layer destroys performance for both $a^2$ (which should be causal towards the end of training) and $b^2$ (which should never be causal). In the MLP block, ablating circuits becomes less detrimental over time for both variables.

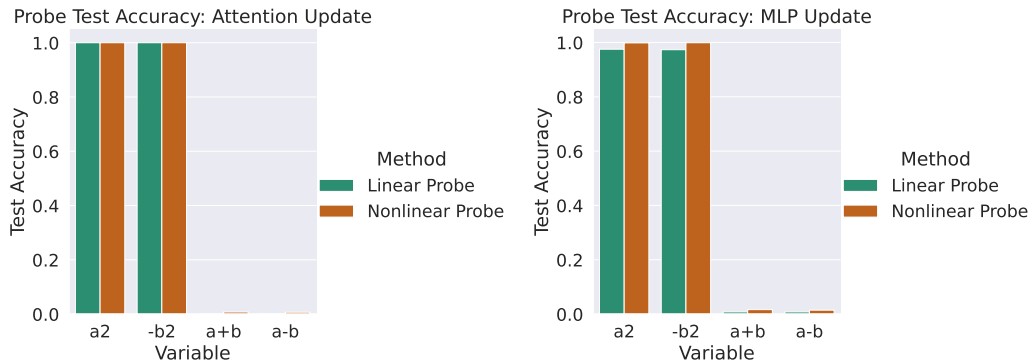

Figure 13: Experiment 1: Results from linear and nonlinear probing on updates from Attention (left) or MLP (right) layers.

to actually compute this variable, so the effect of ablating this set of parameters is hard to predict. Here, we see that it destroys performance.

## Q  Experiments 1-3: Probing on Update Vectors

Circuit probing operates on updates to the residual stream (i.e. vectors generated by MLPs or Attention Layers), rather than the residual stream itself. However, in the main text, we follow common convention and present results from linear and nonlinear probing on the residual stream. In this section we present linear and nonlinear probing results from Experiments 1-3, where we probe the update vectors from MLPs and Attention Layers. In general, we find that probing on either the residual stream or individual vector updates gives extremely similar results. See Figures 13,14,15.

## R  Experiments 1-3: Contrastive Linear Probe

In this section we present an ablation of the circuit probing algorithm, where we optimize a linear probe using a contrastive objective. Formally, we define a linear probe, $P$, which maps from intermediate representations (either residual stream states, Attention updates, or MLP updates) to an embedding dimension $d$. We optimize $P$ using the soft-neighbors loss defined in D. Intuitively, this probe should produce similar embeddings for representations

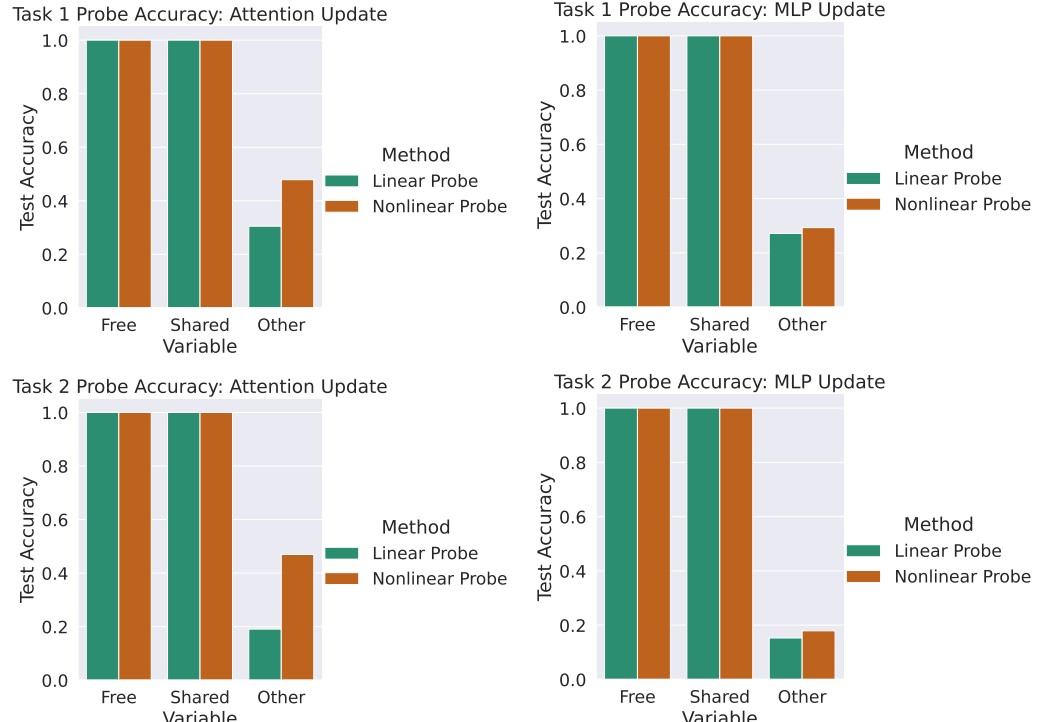

Figure 14: Experiment 2: Results from linear and nonlinear probing on updates from Attention (left) or MLP (right) layers on Task 1 (top) or Task 2 (bottom).

that belong to the same class, and different embeddings for representations that belong to different classes. We evaluate this probe using the same procedure outline in Section 2. For ease of comparison, we fix $d = 113$, which equates the number of parameters in the contrastive probe and the linear probes used in Experiments 1-3. In general, we find that this probe produces fairly poor results (Figure 16). Thus, we cannot attribute the success of circuit probing merely to a different optimization objective.

## S  Experiment 4: Language Prefix Examples

Here, we present several examples of sentence prefixes for subject-verb agreement and reflexive anaphora.

**Subject-Verb Agreement:**

IID: 1-Distractor

1. the farmers that the taxi driver admires (are)
2. the authors behind the assistants (are)
3. the consultant that the skaters like (is)

OOD: 2-Distractors

1. the books by the architects next to the executives (are)
2. the customer that the skaters like and the ministers hate (is)
3. the pilots in front of the dancers to the side of the parents (is)

**Reflexive Anaphora:**

IID: 1-Distractor

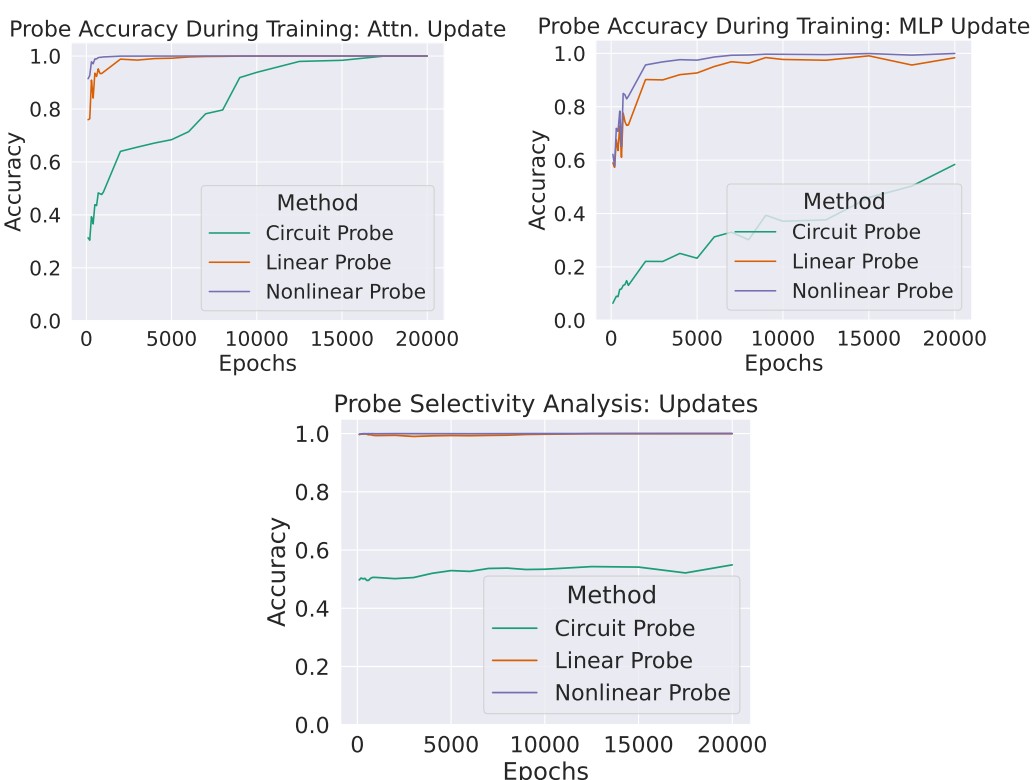

Figure 15: Experiment 3: Results from linear and nonlinear probing on updates from Attention (left) or MLP (right) layers. (Bottom) Selectivity analysis when probing attention layer updates for irrelevant variable.

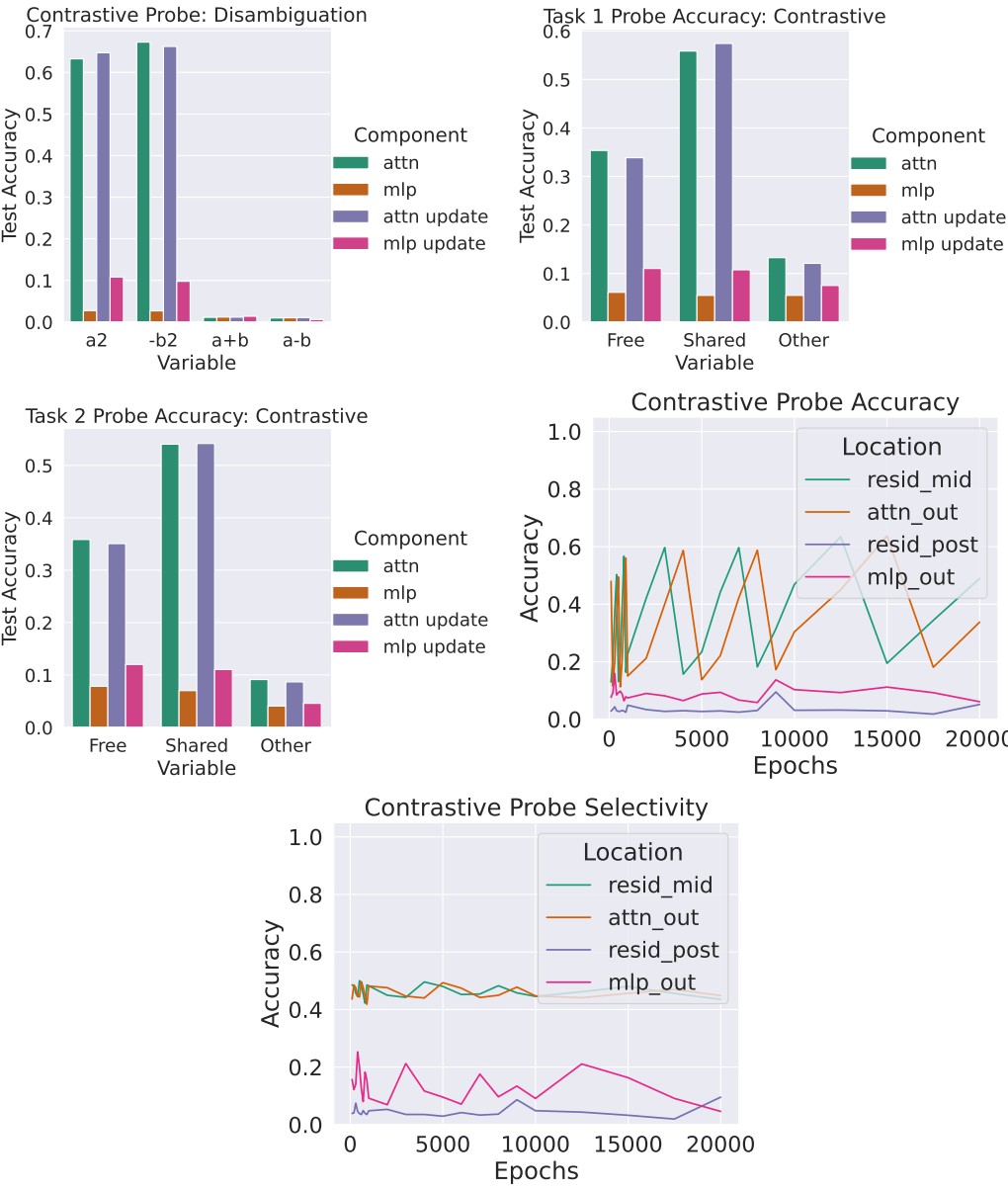

Figure 16: Results from Contrastive Linear Probing in Experiments 1-3. In general, we see that the probe achieves low accuracy across the board. Though it does differentiate between variables that are relevant vs. irrelevant in Experiment 1 (top left) and Experiment 2 (top right, middle left), it fails to do so for Experiment 3 (middle right, bottom).

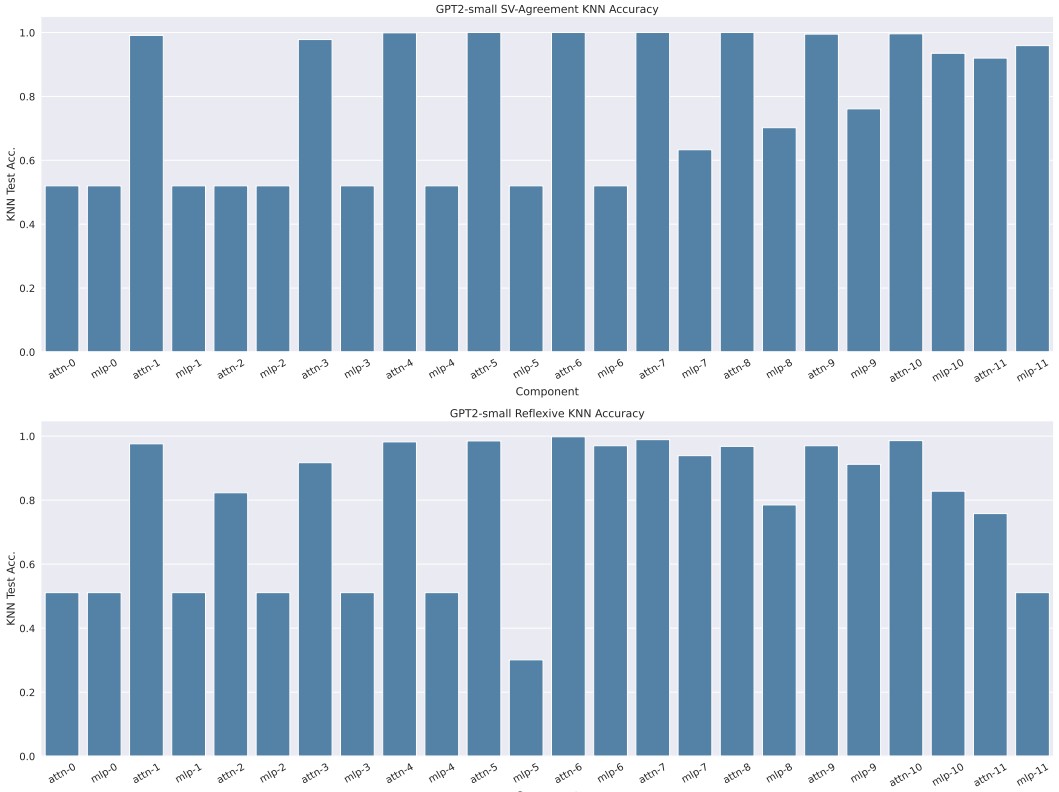

Figure 17: GPT2-Small circuit probing KNN results for subject-verb agreement (Top) and reflexive anaphora (Bottom). We notice that KNN accuracy increases for MLP blocks after layer 6, which is where our causal analysis located the causal circuits for both phenomena.

1. the consultants that the parents loved doubted (themselves)

2. the senators that the taxi drivers hate congratulated (themselves)

3. the mechanics thought the pilot hurt (herself/himself).

OOD: 2-Distractors

1. the surgeon that the chefs love and the parents admire hated (herself/himself)

2. the mechanic knew the banker said the farmer embarrassed (herself/himself)

3. the teachers that the architect hates and the minister admires hurt (themselves)

# T  Experiment 4: GPT2-Small Extended Results

## T.1  Experiment 4: GPT2-Small Language Probing Results

We present circuit probing KNN evaluations for both subject-verb agreement and reflexive anaphora over all model components. Causal analyses indicate that a circuit that is causally implicated in computing syntactic number is located in layer 6's attention block. We note that KNN accuracy increases for every MLP block after layer 6. Because MLP blocks are applied token-wise, this suggests that the information required to decode syntactic number of both subjects and referents is present in the residual stream after this layer but not before. See Figure 17.

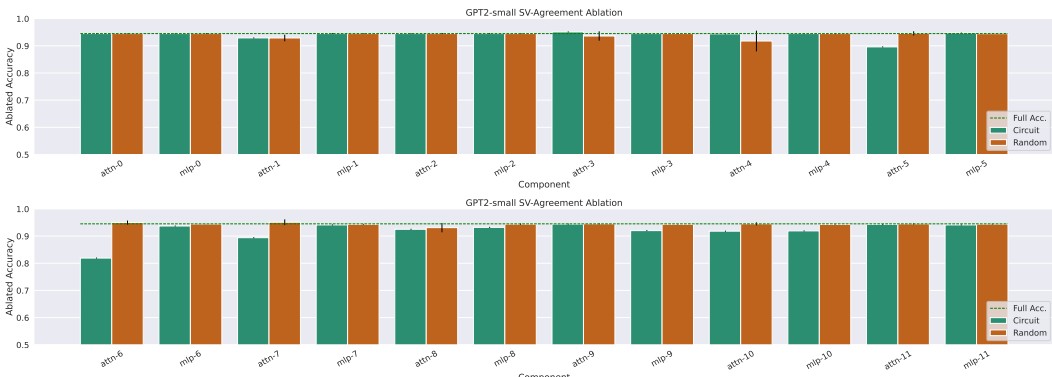

Figure 18: GPT2-Small subject-verb agreement ablation results for every model component. We note that the attention block in layer 6 provides the greatest drop in performance after ablating the discovered circuit.

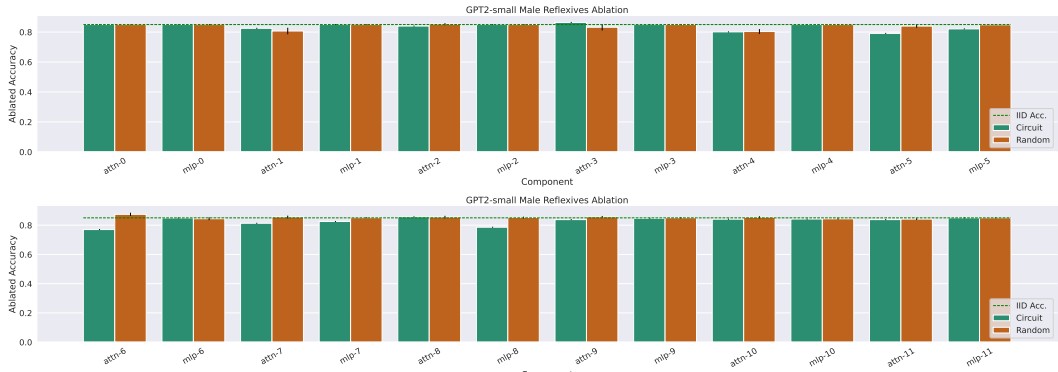

Figure 19: GPT2-Small reflexive anaphora ablation results for every model component, evaluated using the masculine pronoun. We note that the attention block in layer 6 provides the greatest drop in performance after ablating the discovered circuit.

### T.2 Experiment 4: GPT2-Small Full Ablation Results

We present circuit probing ablation results for both subject-verb agreement (See Figure 18) and reflexive anaphora (See Figures 19 and 20) over all model components. In all cases, we note that ablating the circuit in attention block in layer 6 provides the greatest drop in model performance. Randomly ablating subnetworks of the same size does not harm model performance.

## U Experiment 4: GPT2-Medium Results

### U.1 Reflexive Anaphora

We present results analyzing GPT2-Medium's ability to compute the syntactic number of the referent of a reflexive pronoun. We find that the attention block in layer 7 is most causally implicated in this computation. See Figure 21. Ablating the discovered circuits harms model performance, regardless of the pronoun used for evaluation. Ablating random subnetworks of the same size does not harm model performance.

Turning to the reflexive anaphora probing evaluation, we see that the KNN accuracy of circuits trained on MLP blocks increases during and after layer 7. Because MLP blocks operate token-wise, this indicates that the information required to decode the syntactic

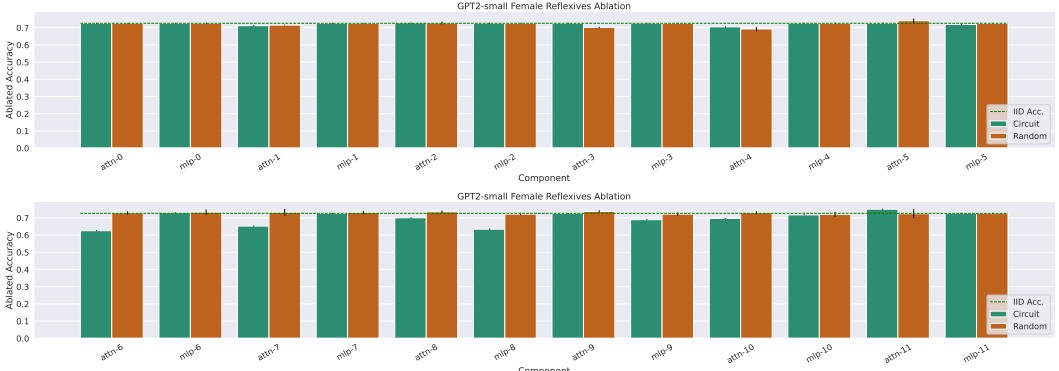

Figure 20: GPT2-Small reflexive anaphora ablation results for every model component, evaluated using the feminine pronoun. We note that the attention block in layer 6 provides the greatest drop in performance after ablating the discovered circuit.

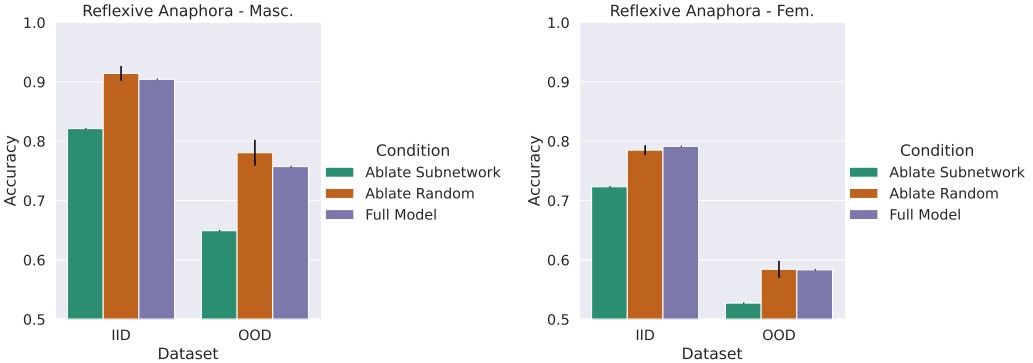

Figure 21: GPT2-Medium reflexive anaphora ablation results for the attention block in layer 7.

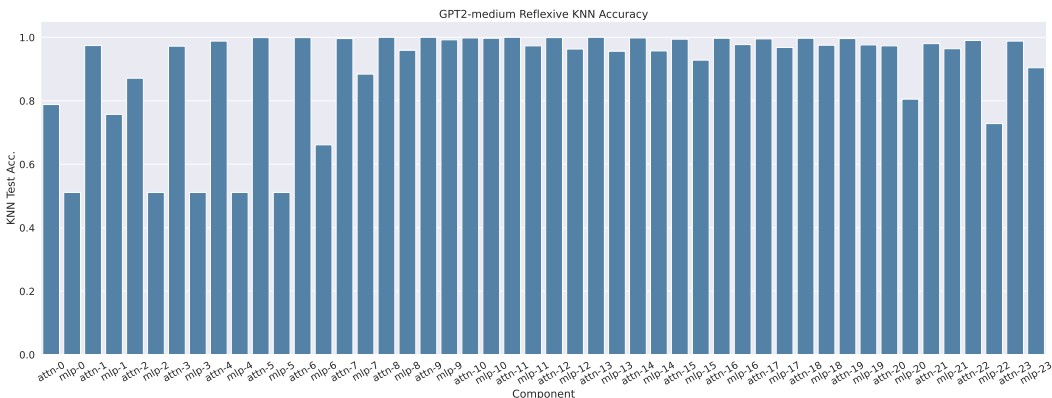

Figure 22: GPT2-Medium reflexive anaphora circuit probing KNN evaluation results across all model components.

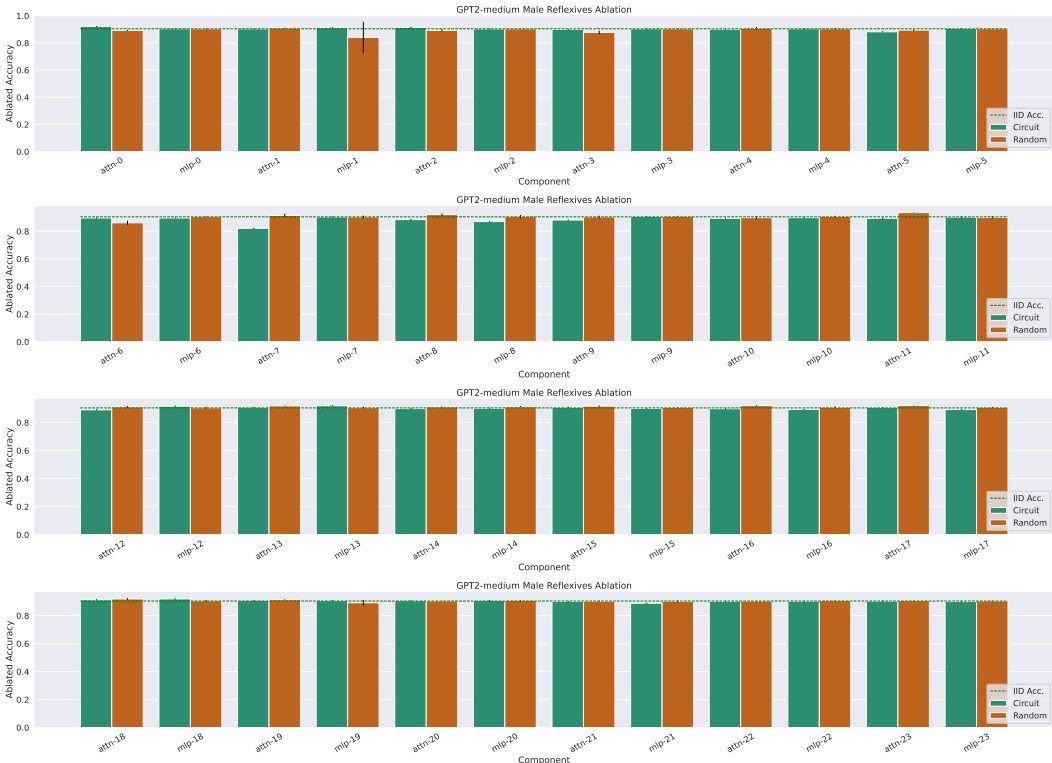

Figure 23: GPT2-Medium reflexive anaphora ablation results across all model components, evaluated using the masculine pronoun. Note that the largest drop in performance due to ablation occurs at the attention block in layer 7.

number of referents is present in the residual stream after this layer, but not before. This strengthens our causal results analysis. See Figure 22.

For completeness, we include ablation results across all model components in Figures 23 and 24.

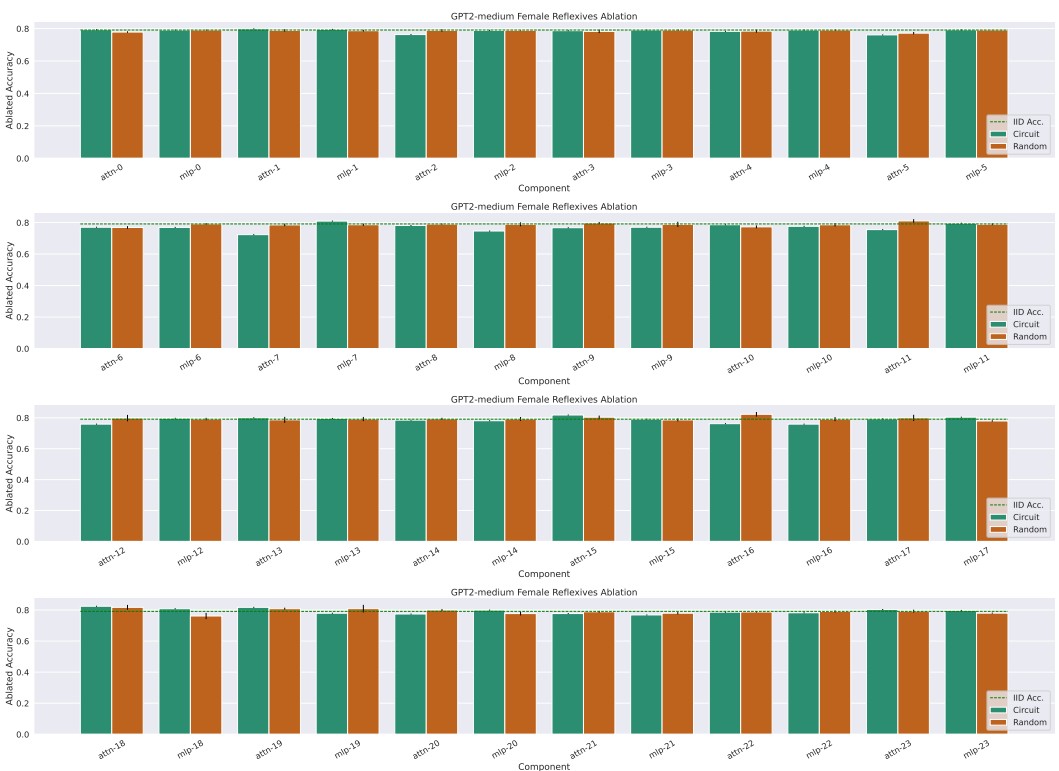

Figure 24: GPT2-Medium reflexive anaphora ablation results across all model components, evaluated using the feminine pronoun. Note that the largest drop in performance due to ablation occurs at the attention block in layer 7.

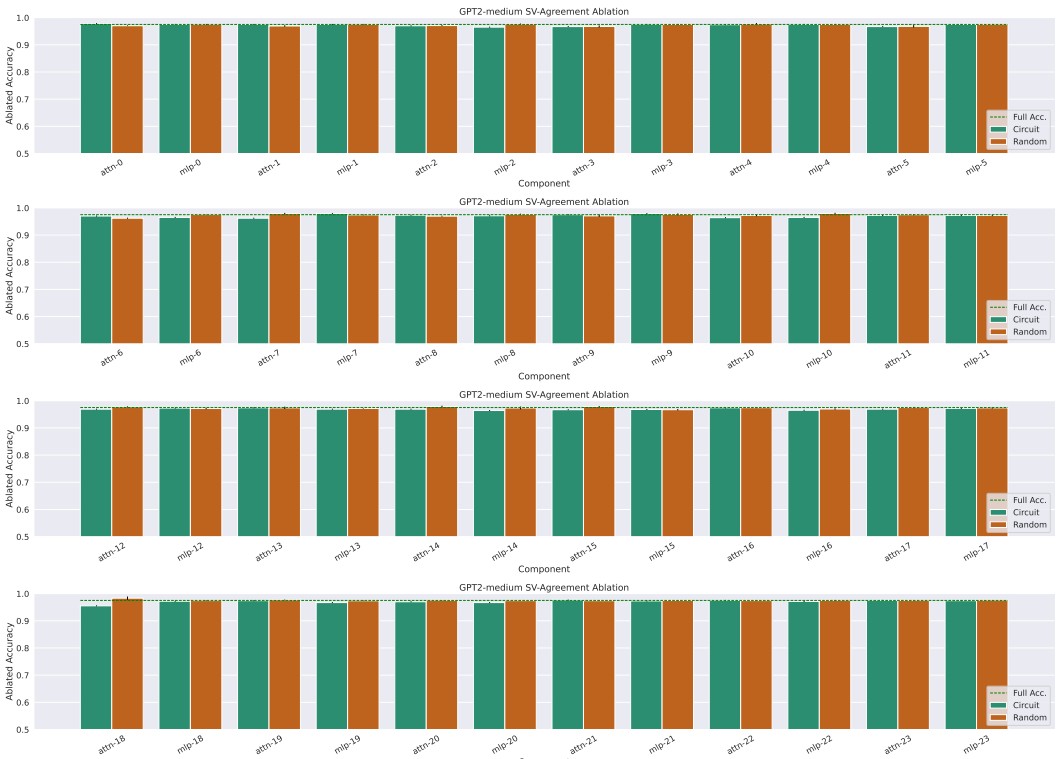

Figure 25: GPT2-Medium subject-verb agreement ablation results across all model components. These results do not implicate any specific circuit in computing the syntactic number of the subject noun.

### U.2   Subject-Verb Agreement

We do not find any particular circuits that drop subject-verb agreement performance substantially when ablated (See Figure 25). This might indicate that multiple circuits across several blocks are redundantly computing the syntactic number of the subject noun.

Turning to the subject-verb agreement probing evaluation, we see that the KNN accuracy of circuits trained on MLP blocks increases after layer 7 (see Figure 26). Because MLP blocks operate token-wise, this might indicate that the information required to decode the syntactic number of referents is present in the residual stream after this layer, but not before. However, our causal analysis does not provide strong evidence of this. From Figure 25, we see a small drop at layer 7's attention block and another at layer 18's attention block.

## V   Experiment 4: Circuit Overlap

Surprisingly, we see that there is very little overlap between the circuits used to compute the syntactic numbers of subjects and referents in GPT2-Small, despite both circuits being present in the same block. See Figure 27.

## W   GPT2-Small Subject-Verb Agreement Qualitative Results

We present qualitative results of ablating the subject-verb agreement circuit discovered by running circuit probing on the attention block in layer 6 of GPT2-Small (See Table 8). We note that the types of tokens predicted by the model qualitatively stay the same before and after ablation. This suggests that we have not destroyed the model by ablating the discovered circuit. Interestingly, we see more tokens that are explicitly consistent with the

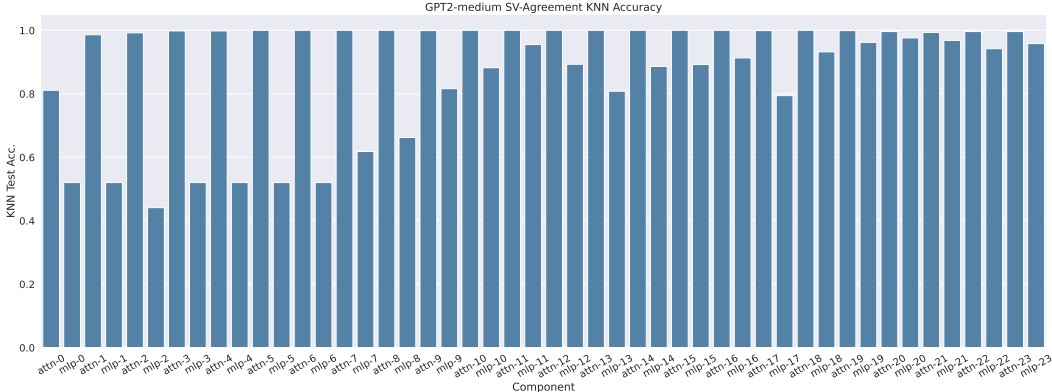

Figure 26: GPT2-Medium subject-verb agreement circuit probing KNN evaluation results. We see KNN performance increase for MLP blocks around layer 7.

syntactic number of the subject before ablation, and fewer after ablation. This provides qualitative evidence that we have indeed ablated a circuit that was responsible for tracking the subject-verb agreement dependency.

## X    Investigating Syntactic Number Feature Extraction

**Goal**    The experiments presented in Section 4.4 demonstrate that linguistic *dependencies* (e.g. the influence of a previous word's syntactic number on main verb prediction) are computed in the middle layers of both GPT2-Small and Medium. However, this leaves open the question: where in the model are the syntactic number features extracted in the first place? In this section, we uncover a circuit that computes syntactic number features and demonstrate that ablating this circuit completely destroys a model's ability to perform subject-verb agreement.

**Task**    We use a subset of the 1-distractor subject-verb agreement dataset from Section 4.4 for this experiment. Specifically, we use the subset of data that includes prepositional phrases (e.g. *the farmer near the parents*). In these sentences, the final word before the main verb is always a non-subject noun.

We run circuit probing on the last token of the non-subject noun in the sentence prefixes to uncover the circuit that computes the syntactic number of the **non-subject noun** (rather than the subject noun). Thus, we are effectively uncovering a syntactic number feature extractor circuit. To ensure that our discovered circuit truly is a general syntactic number feature extractor, we ablate the discovered circuit and test the model's ability to generate syntactically valid next-token predictions using the same evaluation as described in Section 4.4. The experimental logic is that if the discovered circuit is responsible for syntactic number feature extraction *in general*, then ablating it will also destroy a model's ability to extract the syntactic number of the subject noun, rendering the subject-verb agreement task impossible.

**Probing**    We include circuit probing KNN evaluation results for GPT2-Small and Medium Figures 28 and 29. We find that accuracy reaches ceiling after MLP 0 and stays at ceiling until the late stages of the model.

**Causal Analysis**    We expect that ablating the discovered circuit will render the model worse at extracting the syntactic number of all nouns, destroying performance on our subject-verb agreement evaluation. We expect that ablating random subnetworks should not harm performance on this task. We present results from both models (See Figure 30). For both models, we find that the syntactic number is computed in early MLPs (layer 0 for GPT2-Small, layer 1 for GPT2-Medium). Ablating the circuit returned by circuit probing drops

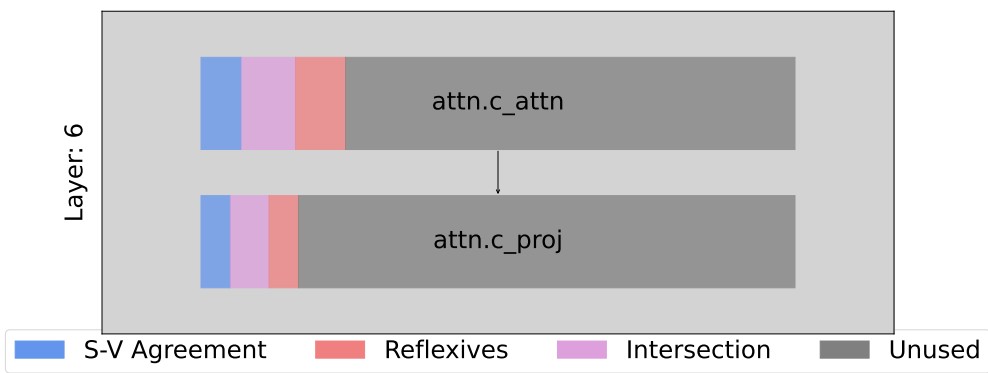

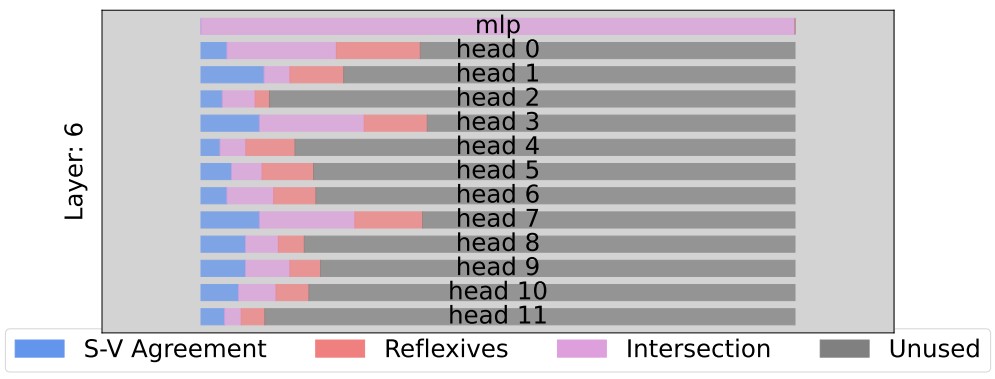

Figure 27: Circuit overlap between syntactic number and reflexive anaphora in GPT2-Small, attention block 6. We see that the discovered circuits are largely distinct. We also note that certain attention heads (0, 3, and 7) appear to be most important in computing syntactic number for both subject nouns and referents.

| Prefix | Original Output | Ablate Attn-6 Output |
|---|---|---|
| The surgeons behind the dancer | 's | 's |
| | , | , |
| | **were** | . |
| | . | and |
| | and | _ |
| | **are** | ) |
| | said | to |
| | _ | in |
| | had | who |
| | **have** | was |
| | to | ). |
| | who | is |
| | in | said |
| | was | ), |
| | ) | **were** |
| The book from the executives | of | of |
| | at | at |
| | | who |
| | , | , |
| | who | and |
| | , | , |
| | . | in |
| | and | . |
| | in | to |
| | that | that |
| | **was** | ) |
| | **is** | on |
| | to | I |
| | themselves | were |
| | on | are |

Table 8: Qualitative examples of the effect of ablating the circuit discovered in GPT2-Small, layer 6. We record the top 15 next-token predictions. Words that are explicitly consistent with the syntactic number of the subject are bolded, words that are inconsistent are underlined.

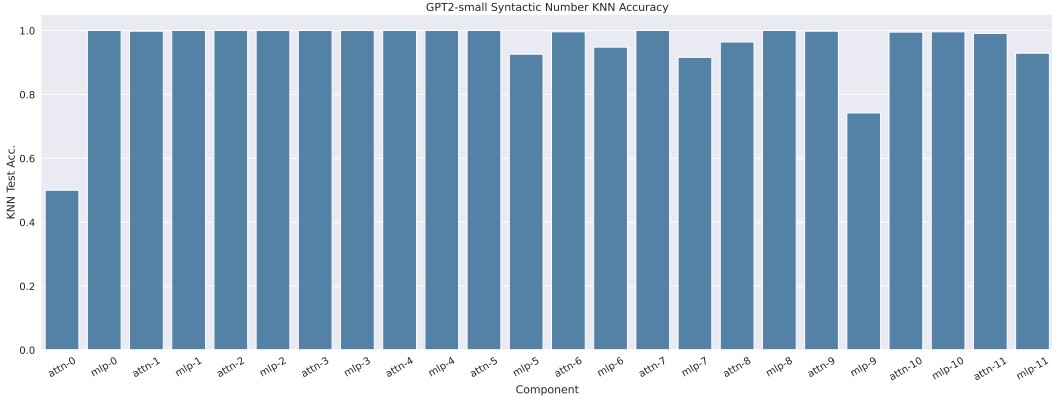

Figure 28: GPT2-Small syntactic number KNN results. We see KNN performance reach ceiling after MLP 0, and stay at ceiling until late in the model.

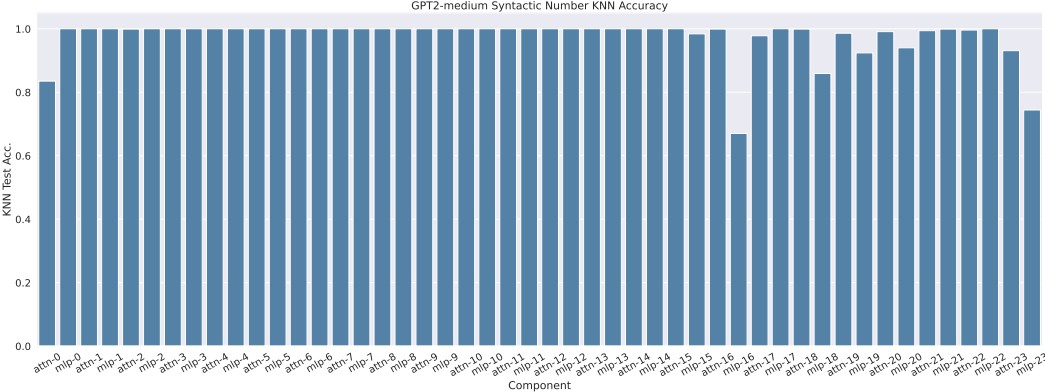

Figure 29: GPT2-Medium syntactic number KNN results. We see KNN performance reach ceiling after MLP 0, and stay at ceiling until late in the model.

subject-verb agreement performance nearly to chance while ablating random subnetworks does not impact model performance. We present results from all layers for GPT2-Small in Figure 31 and for GPT2-Medium in Figure 32. We note that it seems that GPT2-Medium also performs syntactic number extraction in layer 0's MLP.

## Y   GPT2 Linear Probing Results

In this section, we present results generated by linearly probing GPT2-small and medium in all tested conditions (subject-verb agreement, reflexive anaphora, and syntactic number). Across the board, we find that linear probing decodes the syntactic number early, and maintains ceiling performance throughout the network. See Figures 33,34,35. Notably, circuit probing can identify when layers are causally implicated in downstream model performance via subnetwork ablation, whereas linear probing in this manner does not directly provide this information.

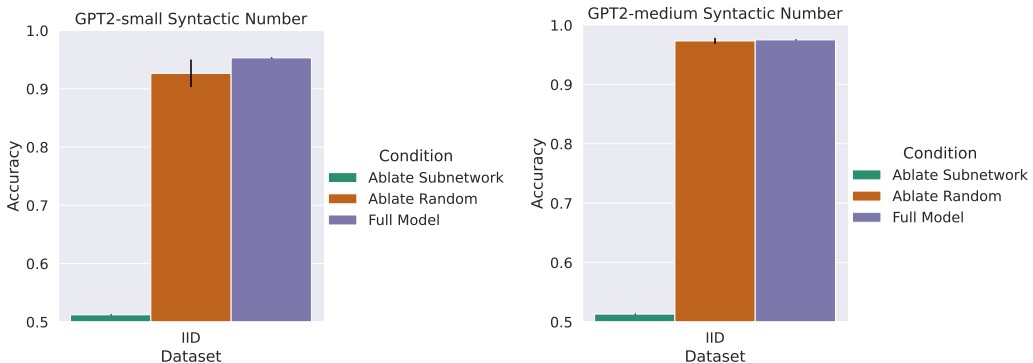

Figure 30: GPT2-Small (left) and Medium (right) ablation results on early MLPs. We find that the circuit that extracts syntactic number features from tokens is computed in early MLPs. Ablating this circuit destroys subject-verb agreement performance while ablating random circuits of the same size does not substantially impact performance.

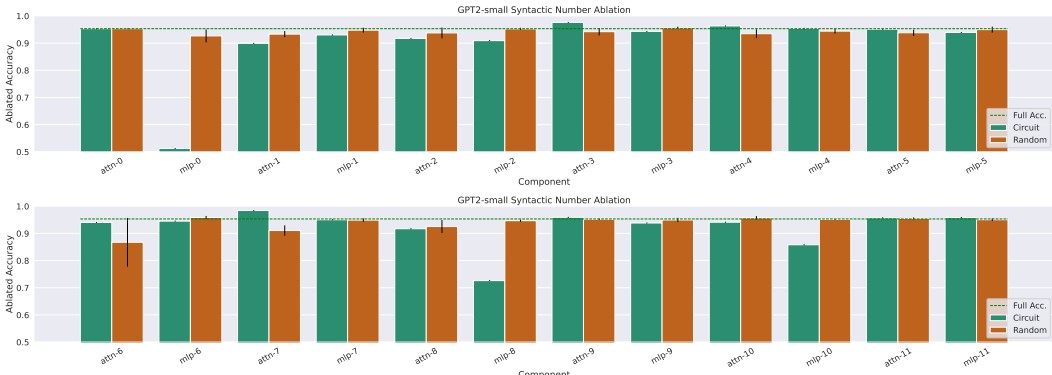

Figure 31: GPT2-Small syntactic number ablation results across all model components. Note that the largest drop in performance due to ablation occurs at the MLP block in layer 0.

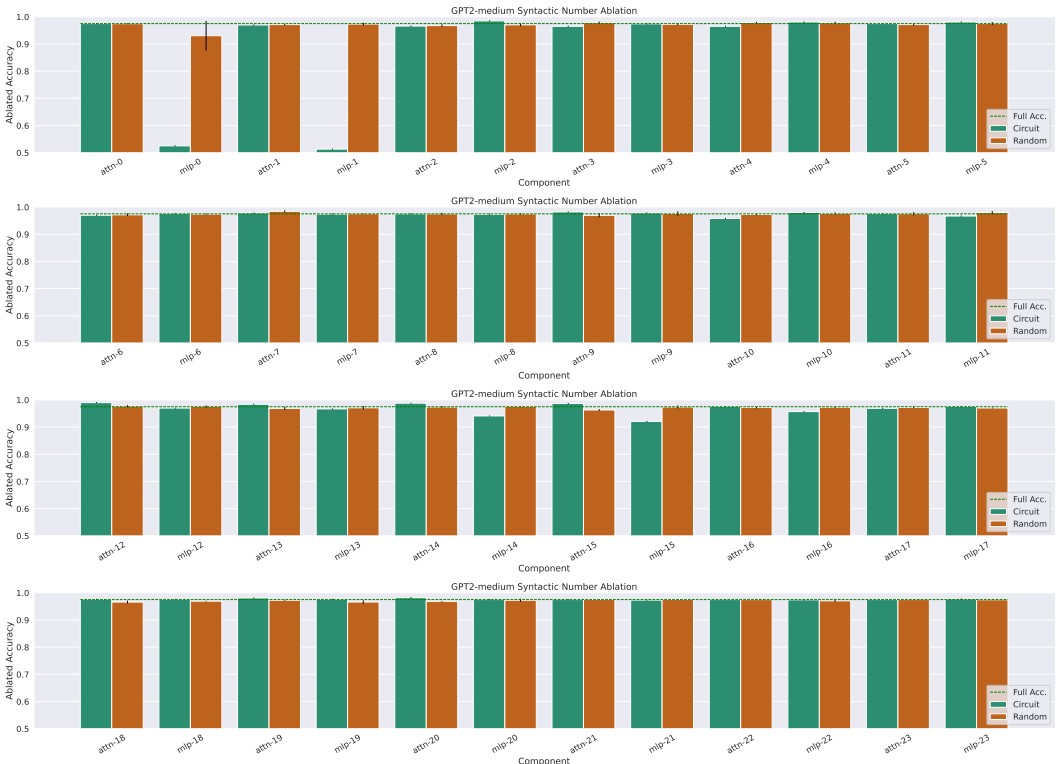

Figure 32: GPT2-Medium syntactic number ablation results across all model components. Note that the largest drop in performance due to ablation occurs at the MLP blocks in layer 1 and layer 0. However, ablating random subnetworks has less of an impact on model performance in layer 1's MLP.

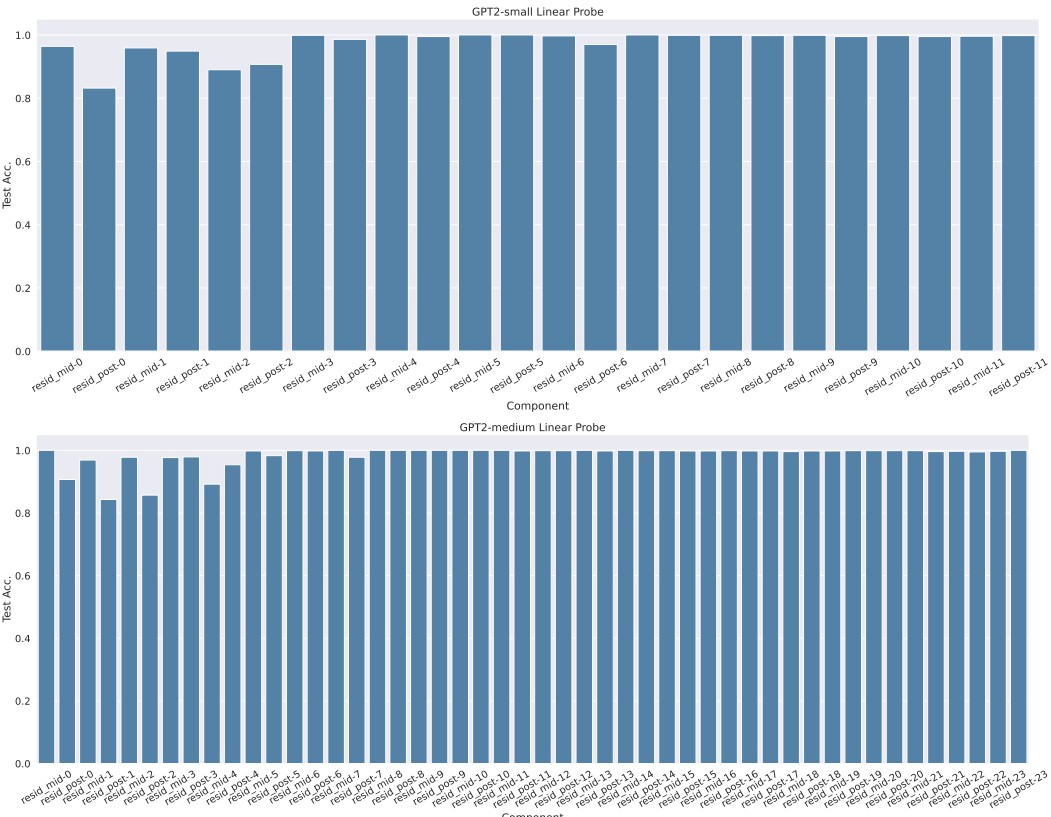

Figure 33: GPT2-small and medium linear probing in Subject-Verb Agreement Task.

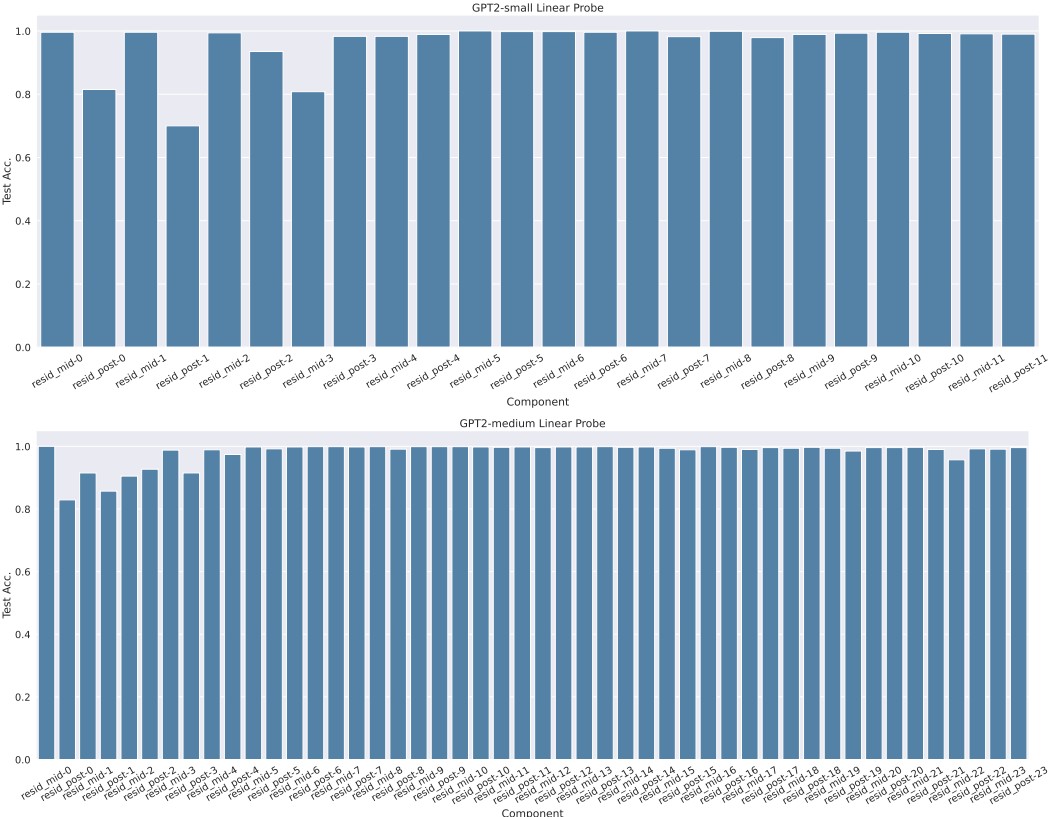

Figure 34: GPT2-small and medium linear probing in Reflexive Anaphora Task.

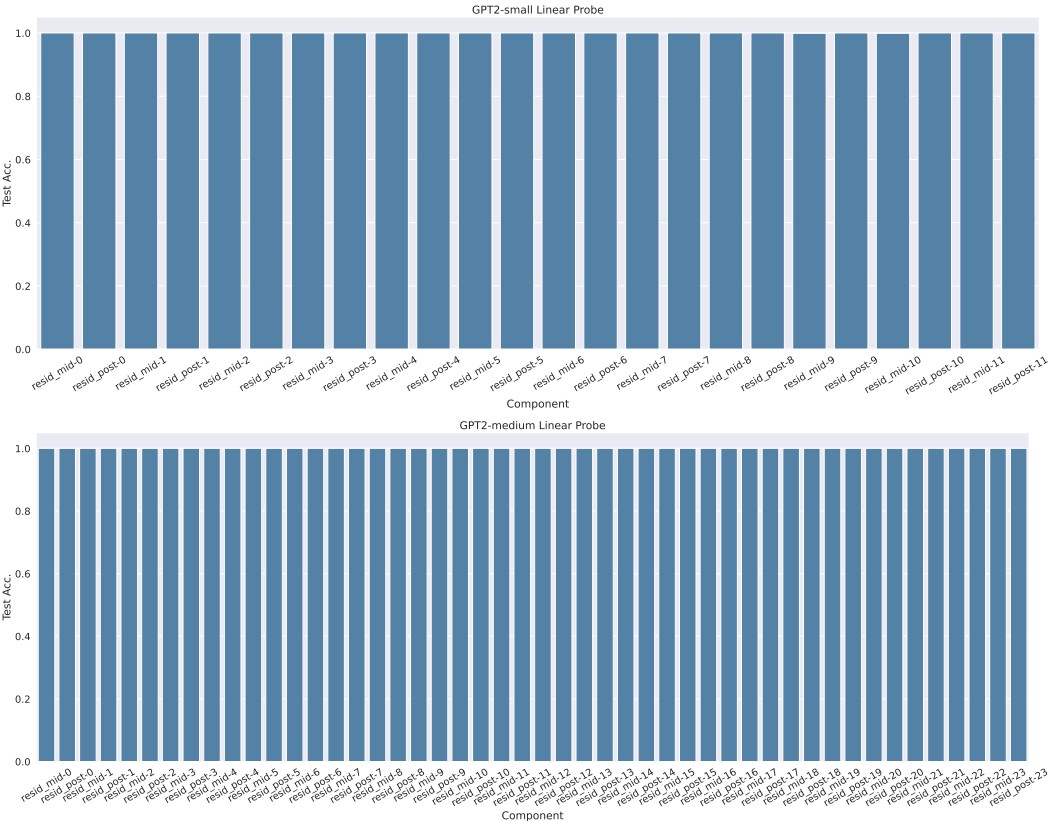

Figure 35: GPT2-small and medium linear probing in Syntactic Number Task.

