# OpenReview forum: "Uncovering Intermediate Variables in Transformers using Circuit Probing"
_colmweb.org/COLM/2024/Conference — COLM_

### Official Review · Reviewer_Dcii · 2024-05-10

**Rating:** 6
**Confidence:** 3
**Ethics Flag:** 1

**Summary:**

This paper presents a novel technique called circuit probing for identifying and analyzing intermediate variables in Transformer models. Circuit probing optimizes a binary mask over model weights to uncover a circuit that computes a hypothesized intermediate variable, then tests if the variable is represented in the model and causally implicated in model behavior, and look for model components whose outputs are partitioned according to the variable. The authors conduct four experiments that demonstrate circuit probing's ability to identify intermediate variables, differentiate between algorithms, and evaluate modularity, outperforming existing probing techniques.

**Questions To Authors:**

1.	What is the scalability of circuit probing when applied to larger, state-of-the-art models used in complex real-world applications? The current study focuses on relatively small models and straightforward tasks like arithmetic.
2.	How robust are the identified circuits to variations in hyperparameters, such as optimization functions, regularization methods, learning rate schedules, etc.?

**Reasons To Accept:**

1. Novelty: Circuit probing is a valuable and innovative contribution to the field of interpretability research, offering a method to understand intermediate variables in Transformer models more effectively than traditional probing methods,
•	Faithful to model's computation than probing or causal abstraction analysis;
•	Not require full causal graph specification like causal abstraction analysis;
•	Uncover development of circuits during training unlike other methods.
2. Diverse Experimentation: The authors validate their methodology with comprehensive experimentation. Applying circuit probing across different tasks, including simple arithmetic tasks and real-world language models, highlights the robustness and versatility of the method.
3. Strong Comparative Analysis: Comparing circuit probing to existing probing methods helps establish its efficacy. The ability to ablate specific circuits and demonstrate causal changes in model behavior is a notable advantage.

**Reasons To Reject:**

1. Accessibility and Clarity: The paper's complexity may limit understanding for researchers not specialized in interpretability or Transformer models. Providing more intuitive explanations and detailed visualizations could enhance accessibility.
2. Generalization: While the experiments are comprehensive, additional work applying circuit probing to different neural network architectures or more complex, real-world tasks would reinforce the paper's claims.
3. Computational Efficiency: Discussing the computational requirements of circuit probing would provide practical insights for broader application. An analysis of scalability could strengthen the paper.

---

> ### Author Rebuttal · Authors · 2024-05-31
>
> Thank you for your helpful comments! See replies to your individual points below:
>
> "Accessibility and Clarity:..."
>
> We are happy to add some higher-level intuitions into the paper, especially in Section 2!
>
> "Generalization:...."
>
> We agree that working on a wider variety of complex models (e.g. vision transformers) is interesting and important. However, this was out of scope for this paper. We analyzed GPT2-small and medium for three particular intermediate variables in a language modeling task (two in the main text, one in the appendix). This was to establish that the method generalizes well to larger architectures trained on much more complex tasks (language modeling).
>
> "Computational Efficiency: ..."
>
> Happy to add an appendix section on this! In short, an optimized version of this method relies on forward passes through the model from the first layer through the Nth, where N is the layer to be probed. The backward pass only needs to be taken at the Nth layer, as that contains the binary masking parameters. In terms of memory requirements, this technique introduces one parameter per masking unit. In this work, we masked at the neuron level, though one could just as easily mask at the weight level or attention head level. In total, this is not very many parameters in comparison with the rest of the model.
>
> "How robust..."
>
> We largely used the default set of hyperparameters suggested by Savarese et al. https://arxiv.org/abs/1912.04427. The one parameter that we change per model is the L0 regularization strength parameter. This is set to initialize the L0 loss in the range of .1-1. Because the maximum (and initial) L0 loss depends on the number of parameters you are masking over, this has to be set on a per model basis: the same L0 regularization strength for a toy model would dominate the loss in GPT2-Medium. In short, this method does not require much hyperparameter tuning at all!

---

> > ### Comment · Reviewer_Dcii · 2024-06-05
> >
> > Thanks for the clarification. I would like to keep my rating the same.

---

### Official Review · Reviewer_XRXi · 2024-05-10

**Rating:** 5
**Confidence:** 3
**Ethics Flag:** 1

**Summary:**

This paper proposes a method for finding subnetworks (called "circuits") in transformers that calculate intermediate variables in a hypothesized causal model. The method is evaluated in terms of two goals. First, if a variable is part of the true causal graph, the method should find a subnetwork that calculates it reliably, and if the variable is not part of the causal graph, the method should return an empty subnetwork. Second, if a subnetwork is found, ablating the subnetwork should harm the performance of the model. The method is compared to probing methods (for the first goal) and causal analysis methods (for the second goal) and generally agrees with existing methods or compares favorably.

**Questions To Authors:**

I have no additional questions

**Reasons To Accept:**

- This paper brings together ideas from pruning, circuit discovery, probing, and causal analysis. Causal analysis methods are designed to test whether neural network representations can be abstracted as high-level variables in a causal model, whereas circuit discovery methods can help characterize the mechanisms the model uses to compute these variables. This kind of combined approach could be a promising tool for understanding transformers at different levels of abstraction.

- The paper includes experiments in a variety of different settings and compares circuit probing to a number of sensible baselines. The experiments suggest that circuit probing can help answer a variety of different interpretability questions.

- The experiments indicate that circuit probing generally agrees with existing methods and in some cases performs better, in the sense that circuit probing agrees more with our expectations. For example, in section 4.3, circuit probing seems to be work than other probing methods as an explanation for "grokking" training dynamics.

**Reasons To Reject:**

- One of the proposed benefits of this approach is that it enables a form of causal analysis, but it is not clear to me how much the method actually reveals about the causal model implicit in the network. In particular, the causal analysis involves ablating the identified circuit, and the paper states that "this ablation is equivalent to asking the counterfactual question 'How does the model’s output change if it does not compute a particular intermediate variable in a particular block?'" However, I do not see how these are equivalent. The question assumes that the chosen subnetwork (a) computes a particular intermediate variable, and (b) does not do anything else. That is, ablating the subnetwork could have other effects on the model that are not related to the intermediate variable. More generally, the paper would be stronger if claims about causality could be stated more formally. For example, existing work on causal analysis (e.g. [1]) uses a more more formal definitions of the sense in which the method can support claims about the causal structure of the neural network. It would be helpful if this paper could be more precise in explaining the type of causal claims that could be supported by the method.

- The paper argues that circuit probing is better than existing probing methods because it is more "faithful" to the original model but it does not provide a clear definition of faithfulness, and it is not clear to me how the experiments demonstrate that circuit probing is more faithful than other methods. In the introduction, faithfulness is defined parenthetically as "it only provides evidence in support of causal variables that are actually represented by the model," but I do not see how there is any ground-truth way of knowing what causal variables the model actually represents. (One possible approach would be to include experiments with networks where the ground truth causal structure is known, e.g. using [Tracr](https://arxiv.org/abs/2301.05062) [2].)

- I do not always find the experiments to be very compelling. Three of the four experiments study one-layer networks trained on very simple tasks, which do not clearly demonstrate that this is an appropriate method for analyzing transformer language models. The fourth experiment studies language modeling tasks, but I am not entirely convinced by the conclusions. Section 4.4 states that "the dependency is computed in layer 6's attention block", but this seems to overstate the result. From Appendix Figure 15, it seems that ablating a subnetwork in this block leads to a larger drop in performance relative to other layers. However, the drop still isn't especially large (e.g. ~95% to ~80% on SV agreement), and ablating circuits in other components also induces a drop in accuracy, albeit not as large. Even if the effect could be entirely localized to this component, it is not clear to me how significant this type of insight is.

- I find the description of the method in section 2 to be somewhat vague. It would be helpful to start with a more formal statement of the problem, and describe the method more formally. One particular point that confused me: When the method is applied to some component C, does this mean that the mask is learned only to the parameters in C? Or is the mask learned for all components prior to C in the computation graph, but using the representations from C to calculate the loss? Is there any justification for one or the other? Without explaining these details, it is difficult to understand how the method works, or even the exact problem it is meant to solve.

- The method involves two ideas: one is to use pruning/subnetwork analysis as a probing method, and the other is to find these subnetworks by optimizing a nearest neighbor loss. These ideas seem to be distinct to me, and I think it would make sense to evaluate these two components of the method individually, that is: (a) the same approach but replacing the nearest neighbor objective with a standard linear probing method; and (b) training a linear probe on the network representations, but using the nearest neighbor objective.

Overall, I think the paper introduces an interesting approach to interpretability, but I think a number of the key points need to be formulated more precisely to be able to determine the types of questions this method is capable of answering.

*References:*

[1] Geiger et al., 2021. Causal Abstractions of Neural Networks.


[2] Lindner et al., 2023. Tracr: Compiled Transformers as a Laboratory for Interpretability.


*Update:* After reading the rebuttal, I still have some doubts about the significance of the method and findings, but the authors have promised to address some of my concerns about clarity and presentation, so I am increasing my score from 4 to 5.

---

> ### Author Rebuttal · Authors · 2024-05-31
>
> Thank you for your detailed and comprehensive and helpful review! Please find responses to your concerns below.
>
> "One of the proposed..."
>
> Relating our method to this sort of counterfactual question was grounded in prior work (https://arxiv.org/pdf/2006.00995), though we will be more precise about the assumptions baked into that relationship (which you correctly identified) in the final submission. With respect to assumption (b), we encourage the discovered subnetworks to be as sparse as possible using L0 regularization, under the assumption that maximally sparse subnetworks should be less susceptible to computing many other causally relevant variables. This point will also be refined.
>
> "More generally..."
>
> As mentioned above, we are happy to include a more formal statement of the counterfactual question, and formally list the assumptions therein.
>
> "The paper argues..."
>
> In experiment 1, we derive ground truth by consensus of all existing methods. In Experiments 2 and 3, we used tasks that contain at least one variable that is explicitly useless in deriving the correct answer. They are (by construction) not causal variables in a model that succeeds at the task. We considered using Tracr, but went with this approach because the weight matrices of Tracr models are artificially constructed, and thus somewhat less realistic.
>
> "I do not always find the..."
>
> We will tone down these claims. However, this subnetwork does appear to be causal. For qualitative evidence of this, see Table 8 in the appendix. In these examples, we observe that ablation pushes verbs that exhibit the correct syntactic number down in the ranking, equal to (or worse than) verbs exhibiting the incorrect syntactic number. Happy to expand on this in the main text!
>
> Scientifically, this contributes toward understanding the internal organization of language models. The subnetworks that we identify as partially computing a syntactic dependency consist of approximately 15% of neurons in a single layer! This localization of intermediate variables is not obvious to us a priori.
>
> "The method involves.."
>
> For suggestion (a), see the response to reviewer UN6h. For suggestion (b): We are happy to include this as a baseline in the work. Specifically: rather than having a linear probe map from embedding dimension to N classes, have it map to D dimensions, where D. Take the D-dimensional representations and compute soft nearest neighbors loss in that space. Is this what you had in mind?

---

> > ### Comment · Reviewer_XRXi · 2024-06-04
> >
> > Thank you to the authors for the detailed response. I think better formalizing the claims about causality and presenting the method more clearly will strengthen the paper, and I am increasing my score accordingly. Regarding my comment about the experiments, thank you for pointing me to the example Appendix Table 8. This example is interesting and worth highlighting, although it doesn't entirely address my concerns---for example, from Fig. 14, a number of other components are also associated with substantial drops in ablated accuracy, so might have similar effects.
> >
> > Regarding my suggestion (b), my point was to separate the nearest neighbor objective from pruning--so training on the same objective, but optimizing a linear projection matrix rather than a parameter mask.

---

### Official Review · Reviewer_UN6h · 2024-05-11

**Rating:** 7
**Confidence:** 3
**Ethics Flag:** 1

**Summary:**

The paper proposes a methodology for assessing the presence of intermediate variables in transformers. Assume you have a dataset of examples, which are labelled with the intermediate variable for a particular phenomenon. Prior work would use this data to train linear/non-linear probing classifiers on the hidden states of the transformer. This paper suggests that we should prune the model with these labels to discover a sub-circuit and that the sub-circuit allows us to discover the intermediate variables more easily. The authors explore this technique in four experimental settings and empirically validate that the discovered circuit components play an important part in the circuit.

**Questions To Authors:**

It seems like the main caption for what should be Figure 2 does not exist?

**Reasons To Accept:**

- The authors study four applications for circuit probing in experimental setting ranging from toy experiments on specially small models (Experiments 1 and 2), a study of training dynamics (Experiments 3), and interpretability on a pre-trained language model with millions of parameters (Experiment 4)
- The paper proposes imaginative new experimental settings (Experiments 1 and 2) that illustrate the role of intermediate variables and how they can lead to better understanding
- The empirical analysis is very thorough and leverages a number of related works (amnesic probing, causal abstractions, linear probing, non-linear probing)

**Reasons To Reject:**

- The main method could be better motivated, presented and ablated. How important  is the contrastive similarity loss? Would the method still work with a standard cross-entropy loss over the intermediate labels? How important is pruning as opposed to standard fine-tuning or PEFT? What if we performed prior probing methods on residual stream updates as opposed to hidden states?
- The causal analysis in Experiments 2-4 is weak, as it relies too heavily on performance drops under targeted ablations compared to random ablation. Rather, the authors should consider activation patching and more targeted interventions to demonstrate causality.
- While the paper makes a good case for the limitations of linear and non-linear probing in Experiments 1-3, Experiment 4 seems to lack these simple baselines.

---

> ### Author Rebuttal · Authors · 2024-05-31
>
> Thank you for your feedback on our work! We are happy to address these comments before publication. Please find specific responses to your comments below:
>
> “ The main method ... intermediate labels?”
>
> Great question! The contrastive similarity loss is in some sense a natural choice for our goal, which is to uncover the set of weights required to compute a particular intermediate variable used by a transformer. This loss allows us to compute a loss value that just takes in intermediate representations, rather than logits. Consider using a CE loss for the same problem. Whatever subnetwork we discover will not produce logits, and so one cannot compute a CE loss directly One could train a classifier over the representations generated by these subnetworks (https://arxiv.org/abs/2104.03514). However, this recapitulates the problems with using a linear probe on our representations in the first place!  We are happy to elaborate on this in the main text.
>
> “How important is pruning as opposed to standard fine-tuning or PEFT?”
>
> I’m actually not quite sure what you mean here. Our goal is to uncover the set of model weights responsible for computing a particular intermediate variable in a transformer. I’d love to hear what you are thinking though!
>
> " What if we performed prior probing methods...”
>
> We {nonlinearly, linearly, amnesic} probe after the attention block and after the MLP block. The difference in performance of a standard probe from one to another can only be accounted for by the residual stream updates produced by either block, and so we do not anticipate that this would change our results. We ran linear probing on the update vectors for experiment 3, and found no change in results. We will repeat this for all of our experiments and include this in the appendix.
>
> “The causal analysis...”
>
> We agree that counterfactual interventions provide stronger characterizations of the causal role of the intermediate variables that we discover, and this is a major goal of this set of projects. To do so, one must directly tie the representations generated by subnetworks to representations generated by complete model components. We hope to characterize this relationship in future work but thought it out of scope for this paper. However, we note that ablation has proven useful for interpretability (https://arxiv.org/abs/2006.00995, https://arxiv.org/pdf/2306.03819).
>
> "While the paper ..."
> We are happy to include these results in the final submission for added context!

---

> > ### Comment · Reviewer_UN6h · 2024-06-06
> > **Reviewer response**
> >
> > Thank you for the clear response! Since the authors committed to including the additional baselines in the final version, I will increase my score. Along with reviewer XRXi I would encourage the authors to add clarifications regarding the claims of causality and add a limitations section.
> >
> > A quick clarification:
> >
> > > Computing the CE loss
> > When I suggested the alternative method of training with CE loss instead of a contrastive loss, I was thinking of initializing a new classification head to compute the CE loss. However, I agree that the lack of newly initialized parameters is an advantage of the contrastive loss!

---

### Decision · Program_Chairs · 2024-07-10

**Decision:**

Accept

**Comment:**

The reviewers are all supportive of this paper, which presents and evaluates a new probing method for understanding the computation of a latent within a large neural network. The new method jointly learns a binary mask, identifying a sparse subcircuit that computes the probed variable within the model.

Strengths:

Reviewers agree that the paper is interesting, bringing together ideas from both probing and circuit identification methods, and testing the method on a diversity of settings and networks, including some interesting experiment settings that pull apart the latent variables that emerge within cleverly chosen synthetic problems that one reviewer described as "imaginative".  The new method and the new experiments are meaningful contributions to the literature and they are well-grounded in previous ideas.

Weaknesses:

The reviewers all agree that the presentation of the method is not very clear: Section 2 is a very informal description of the method that reviewers described as "vague", whose "complexity may limit understanding" that "could be better presented". In short, there is no crisp description of the method in the main paper, and many of the key details are left to the appendix or to the supplied code.

This is a significant issue: a clear presentation of the method is especially important if the new method hopes to be understood and adopted by other researchers - it would be a shame if the promising new method were not adopted due to lack of clarity of presentation. While the review consensus is that the paper is above the bar, I would strongly encourage the authors to spend some time for a careful revision of section 2 to improve the accessibility and usefulness of the paper to a  broader set of readers.

Some of the reviewers also found that the causality claims were insufficiently supported by results, and the authors have offered to tone down, formalize and clarify the claims in this area.

[comment from the PCs] It's critical the authors revise their paper following the suggestions of the AC, especially to address clarity issues.